# Information Gain-based Policy Optimization: A Simple and Effective Approach for Multi-Turn Search Agents

**Guoqing Wang**[1*], **Sunhao Dai**[1,2*†], **Guangze Ye**[3*], **Zeyu Gan**[2], **Wei Yao**[2], **Yong Deng**[1]
**Xiaofeng Wu**[1], **Zhenzhe Ying**[1]
[1]Venus Team, Ant Group, [2]Renmin University of China, [3]Individual Author
`{guoqingwang905, sunhaodai, guangzeye98}@gmail.com,`
`zygan@ruc.edu.cn, wei.yao@ruc.edu.cn,`
`{dengyong.deng, congyu.wxf, zhenzhe.yzz}@antgroup.com`

## Abstract

Large language model (LLM)-based agents are increasingly trained with reinforcement learning (RL) to enhance their ability to interact with external environments through tool use, particularly in search-based settings that require multi-turn reasoning and information acquisition. However, existing approaches typically rely on outcome-based rewards that are provided only after generating the final answer. This reward sparsity becomes particularly problematic in multi-turn settings, where long trajectories exacerbate three critical issues: (i) advantage collapse, where all rollouts receive identical rewards and provide no useful learning signals; (ii) lack of fine-grained credit assignment, where the correctness of intermediate turns is obscured, especially in long-horizon tasks; and (iii) poor sample efficiency, where each rollout yields only a single outcome signal, leading to low data utilization. In this paper, we propose Information Gain-based Policy Optimization (IGPO), a simple yet effective RL framework that provides dense and intrinsic supervision for multi-turn agent training. IGPO models each interaction turn as an incremental process of acquiring information about the ground truth, and defines turn-level rewards as the marginal ground-truth confidence gain across consecutive turns. Unlike prior process-level reward approaches that rely on external reward models, manual process annotations, or costly Monte Carlo estimation, IGPO obtains dense intrinsic rewards directly from the model's own belief updates. These intrinsic turn-level rewards are combined with outcome-level supervision to form dense reward signals. Extensive experiments on both in-domain and out-of-domain benchmarks demonstrate that IGPO consistently outperforms strong baselines in multi-turn scenarios, achieving higher accuracy and improved data efficiency. Our code is available at `https://github.com/GuoqingWang1/IGPO`.

## 1 Introduction

Large language model (LLM)–based agents are increasingly equipped with the ability to interact with external environments through tool use (Zhang et al., 2026a; Huang et al., 2025; Li et al., 2025c), a capability often regarded as a critical step toward building general-purpose autonomous intelligent systems (Gutierrez et al., 2023; Qu et al., 2025). One of the most prominent application scenarios is *agentic search*, where an agent invokes tools such as web search (Zhang et al., 2025; Qi et al., 2025) to access up-to-date, large-scale knowledge that substantially enhances its ability to solve complex, knowledge-intensive tasks (Ning et al., 2025). Through iterative interaction with the external environment via such tools, search agents can gradually acquire missing information and refine their reasoning trajectories toward solving the target query.

---

[*]Equal contributions.
[†]Corresponding author.

To equip general-purpose LLMs with such agentic search capabilities, early efforts primarily relied on prompt-based workflows (Li et al., 2025b; Wang et al., 2024b; Zheng et al., 2024), which allowed tool use without additional training but often suffered from poor generalization. More recent studies have explored supervised fine-tuning (SFT) (Wang et al., 2024) and reinforcement learning (RL) (Jin et al.; Song et al., 2025a; Zheng et al., 2025b) to explicitly incentivize tool use, achieving markedly better performance. In particular, Group Relative Policy Optimization (GRPO) (Shao et al., 2024)–style methods have emerged as the dominant approach for training agentic LLMs. In this paradigm, a group of rollouts is generated for each query under the current policy, and outcome-based rewards, typically defined by the correctness of the final answer against the ground truth, are used to construct group-relative advantages that drive policy optimization.

Despite their simplicity and effectiveness in relatively easy tasks, outcome rewards suffer from inherent sparsity (Zhang et al., 2026b), providing supervision exclusively at the final answer. This limitation is particularly detrimental in multi-turn agentic settings, where long trajectories exacerbate three critical issues. **First**, outcome-only rewards frequently lead to *advantage collapse*: when intra-group rollouts yield identical answers (e.g., uniformly correct or incorrect), they receive identical rewards, yielding zero group-relative advantages and no gradient signal.

As shown in Figure 1, a substantial portion of training iterations suffer from this issue, especially for smaller models, which struggle more with complex queries. **Second**, outcome supervision *fails to provide fine-grained credit assignment*. In multi-turn scenarios, later turns are tightly dependent on earlier ones: the action of the current turn may be correct but rendered useless by prior mistakes, or conversely, early successes may be negated by subsequent errors. Thus, distinguishing the correctness of intermediate turns is essential in this scenario, but outcome rewards tend to blur such specific correctness. **Third**, outcome reward sparsity results in

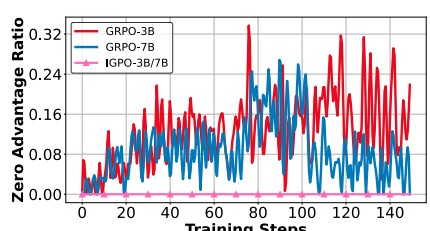

Figure 1: Proportion of zero-advantage groups during training—IGPO vs. GRPO on Qwen2.5-3B/7B-Instruct.

*poor sample efficiency*. Since the entire trajectory receives only a single terminal signal, dense intermediate information is wasted, requiring significantly more samples to learn effective policies.

Recent approaches have attempted to mitigate these issues by introducing process rewards. One line of work leverages external oracle knowledge or reward models to judge intermediate steps (Wang et al., 2025; Feng et al., 2026), but this strategy is costly and risks introducing bias. Another line relies on Monte Carlo simulations to estimate step values (Wang et al., 2024a; Zuo et al., 2026; Zhang et al., 2026b), yet these methods suffer from high variance unless a large number of samples are collected. Overall, both directions face challenges in scalability and fail to provide simple and stable supervision, underscoring the need for an intrinsic and reliable process reward design.

To address these challenges, we propose Information-Gain-based Policy Optimization (IGPO), a simple yet effective RL framework that provides stable and intrinsic supervision for multi-turn search agent training. The key intuition is to model each agent–environment interaction turn as an incremental process of acquiring information about the ground truth. Specifically, at every intermediate turn, IGPO computes the model's ground-truth confidence and defines the turn-level reward as the corresponding ground-truth confidence gain relative to the previous state. This information gain reward offers ground-truth-aware feedback at intermediate turns, in contrast to outcome rewards that only supervise the final answer. However, because outcome rewards are still necessary to anchor learning to the final objective, IGPO incorporates them alongside the turn-level rewards to construct dense reward signals for each rollout. To further stabilize training, we normalize the information gain rewards and outcome rewards separately within groups and propagate them with discounted accumulation, enabling the computation of turn-level discounted returns that capture long-horizon dependencies. Finally, IGPO optimizes the policy with a GRPO-style surrogate objective, replacing rollout-level advantages with our turn-level discounted returns. Additionally, we introduce a vectorized implementation to minimize the computational overhead of information gain rewards.

To evaluate the effectiveness of IGPO, we conduct extensive experiments on both in-domain and out-of-domain benchmarks with search-based agents. Results show that IGPO consistently outperforms strong baselines, delivering substantial gains in both answer accuracy and sample efficiency. Our main contributions can be summarized as follows: (1) We analyze the phenomenon of advantage collapse in outcome reward–based optimization, and reveal the inefficiency of existing process-

level rewards due to reliance on external knowledge or high-variance estimation. (2) We propose IGPO, a simple yet effective policy optimization framework that leverages turn-level information gain to provide dense, ground-truth-aware supervision with negligible computational overhead. (3) Comprehensive experiments demonstrate that IGPO outperforms strong baselines across multiple benchmarks and significantly improves sample efficiency, especially for smaller models.

## 2 PRELIMINARIES

### 2.1 TASK FORMULATION

Let $\mathcal{D} = \{(q, a)\}$ denote a dataset of question–answer pairs, and let $\mathcal{E}$ represent an external tool (e.g., a web search engine). Given a question $q$, the goal of the agent is to solve it by iteratively interacting with the environment through tool $\mathcal{E}$. A rollout trajectory is represented at the turn level as $o = (\tau_1, \tau_2, \ldots, \tau_T)$, where $T$ is the total number of interaction turns in this rollout. The final turn $\tau_T$ is the *answer turn*, which outputs a rationale-then-answer sequence, while all previous turns involve reasoning and tool use. Specifically, for $t < T$, $\tau_t$ is defined as a triple consisting of `[think]`, `[tool call]`, and `[tool response]`. The `[think]` step encourages the agent to reason before acting, with reasoning content wrapped in `<think></think>` tags following the DeepSeek-R1 setting (Guo et al., 2025). The `[tool call]` step invokes the external tool $\mathcal{E}$ through a structured request wrapped in `<tool_call></tool_call>` tags, and the `[tool response]` step returns structured outputs from the environment in `<tool_response></tool_response>` tags. In the final turn, after reasoning, the agent generates its answer within `<answer></answer>` tags, and this content is extracted as the trajectory's final prediction $\hat{a}$. This agent–environment interaction is illustrated at the bottom of Figure 2.

To distinguish model-generated tokens from environment-provided tool responses, we separately define the generated-token sequence and its conditioning context. Let $u = (u_1, u_2, \ldots, u_M)$ denote the sequence of model-generated decision tokens in rollout $o$, where $M$ is the total number of model-generated decision tokens. These tokens include reasoning tokens, tool-call tokens, and answer tokens. Importantly, $u$ does not include tool-response tokens from the environment. For each generated token $u_m$, let $c_m$ denote the complete interaction history immediately before generating $u_m$. The context $c_m$ includes the original query, all previously generated decision tokens, and all tool responses observed so far. Thus, tool-response tokens are used as conditioning context but are never treated as generated action tokens. This distinction allows us to optimize the policy over model-generated tokens while later assigning rewards at the turn level.

### 2.2 AGENTIC REINFORCEMENT LEARNING PIPELINE

**Policy Optimization.** Agentic RL typically optimizes the agent policy $\pi_\theta$ with policy-gradient methods. A representative critic-free method is *Group Relative Policy Optimization* (GRPO) (Shao et al., 2024), which normalizes rewards within each sampled rollout group. Let $\pi_\theta^{\mathcal{E}}(\cdot \mid q)$ denote the rollout distribution induced by policy $\pi_\theta$ when interacting with tool $\mathcal{E}$. Given $(q, a) \sim \mathcal{D}$, a group of $G$ rollouts $\{o_i\}_{i=1}^G$ is sampled from the old policy $\pi_{\theta_{\text{old}}}^{\mathcal{E}}(\cdot \mid q)$. For rollout $o_i$, we write $o_i = (\tau_{i,1}, \tau_{i,2}, \ldots, \tau_{i,T_i})$, where $T_i$ is the number of turns. Let $u_i = (u_{i,1}, u_{i,2}, \ldots, u_{i,M_i})$ denote the sequence of model-generated decision tokens in $o_i$, where $M_i$ is the number of such tokens. For each token $u_{i,m}$, let $c_{i,m}$ be the complete interaction history before generating it, including the query, previous generated tokens, and previous tool responses. Each rollout receives an outcome reward $r_i^{\text{O}}$. GRPO computes the group-relative advantage as $\widehat{A}_i = \frac{r_i^{\text{O}} - \mu_{\text{O}}}{\sigma_{\text{O}}}$, where $\mu_{\text{O}}$ and $\sigma_{\text{O}}$ are the mean and standard deviation of $\{r_i^{\text{O}}\}_{i=1}^G$. For each generated token $u_{i,m}$, the token-level importance ratio is defined as $\rho_{i,m}(\theta) = \frac{\pi_\theta(u_{i,m} \mid c_{i,m})}{\pi_{\theta_{\text{old}}}(u_{i,m} \mid c_{i,m})}$, where $1 \le m \le M_i$. The GRPO objective is

$$
\mathcal{J}_{\text{GRPO}}(\theta) = \mathbb{E}_{(q,a) \sim \mathcal{D}, \{o_i\}_{i=1}^G \sim \pi_{\theta_{\text{old}}}^{\mathcal{E}}(\cdot \mid q)} \left[ \frac{1}{G} \sum_{i=1}^G \frac{1}{M_i} \sum_{m=1}^{M_i} \left( \min\left( \rho_{i,m}(\theta) \widehat{A}_i, \right. \right. \right.
$$

$$
\left. \left. \left. \text{clip}(\rho_{i,m}(\theta), 1 - \epsilon, 1 + \epsilon) \widehat{A}_i \right) - \beta \, \mathbb{D}_{\text{KL}}(\pi_\theta \, \| \, \pi_{\text{ref}}) \right) \right],
$$

(1)

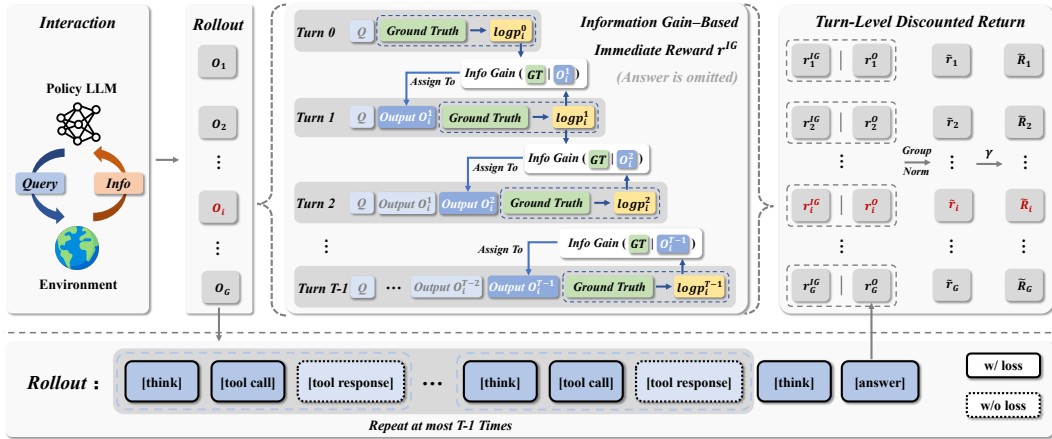

Figure 2: The training pipeline of IGPO. (Upper) Turn-level information gain rewards are computed by measuring changes in ground-truth confidence and combined with the outcome reward to derive discounted returns. (Lower) Each rollout contains multiple interaction turns, where each turn includes a reasoning step, a tool call, and the returned tool response, followed by a final answer turn.

where $\epsilon$ is the clipping ratio, and $\beta$ controls the token-level KL penalty relative to $\pi_{\text{ref}}$ (Shao et al., 2024). Only model-generated decision tokens are optimized; tool responses are loss-masked.

**Reward.** During training, the agent receives a scalar reward $r_i^{\text{O}}$ for each rollout $o_i$, which provides the optimization signal. Prior work typically combines an outcome reward with a format penalty:

$$r_i^{\text{O}} = \begin{cases} \text{F1}(\hat{a}_i, a) = \frac{2\,|\hat{a}_i \cap a|}{|\hat{a}_i| + |a|} \in [0, 1], & \text{if the output is in valid format,} \\ \lambda_{\text{fmt}}, & \text{otherwise,} \end{cases} \tag{2}$$

where $\hat{a}_i$ is the predicted answer of rollout $o_i$, $a$ is the ground truth shared by all rollouts for the same query $q$, and $\text{F1}(\hat{a}_i, a) \in [0, 1]$ denotes the word-level F1 score. Here, $|\hat{a}_i \cap a|$ denotes the word-level overlap between prediction and ground truth. If the output violates the required schema (e.g., missing tags or malformed JSON), a negative constant $\lambda_{\text{fmt}} < 0$ is assigned. Thus, the F1 score provides a correctness signal, while the format penalty enforces structural validity.

## 3 INFORMATION GAIN-BASED POLICY OPTIMIZATION

In this section, we first illustrate our motivation and then provide a detailed introduction to our proposed information gain-based policy optimization, whose overall framework is shown in Figure 2.

### 3.1 MOTIVATION

While outcome-based RL is effective in single-turn tasks, extending it to multi-turn agentic settings faces three critical limitations. First, standard GRPO leads to *advantage collapse*. In the standard framework (Eq. 1), each rollout $o_i$ receives a scalar reward derived solely from the final answer. For complex (or trivial) queries, rollouts often yield identical outcomes (uniformly zero or one), causing group-relative advantages to vanish and providing no valid gradient signal. Second, outcome-only supervision *lacks fine-grained credit assignment*. In multi-turn scenarios, later decisions strictly depend on earlier ones: a tool call may be conceptually correct but rendered useless by prior errors, or conversely, valid reasoning may be overshadowed by subsequent mistakes. Outcome rewards obscure these dependencies, failing to distinguish productive steps from invalid ones. Third, outcome reward sparsity results in *poor sample efficiency*. By relying solely on a single terminal signal, the dense semantic information embedded in intermediate reasoning and tool interactions is wasted, necessitating significantly more samples to learn effective policies.

To mitigate this issue, we introduce Information-Gain-based Policy Optimization (IGPO). The key idea is to exploit the multi-turn structure of agentic rollouts and treat each turn as an opportunity to acquire additional evidence toward the ground truth. At every intermediate turn, IGPO measures the change in ground-truth confidence before and after the current turn and uses it as the

turn-level reward. By rewarding turn-level information gain, IGPO supplies denser and more fine-grained supervision, especially at early training stages. We further present a theoretical analysis in Appendix A, which intuitively explains why IGPO effectively addresses the limitations of sparse outcome rewards in multi-turn scenarios. Since the information gain is defined with respect to the ground-truth answer and computed under teacher forcing, it provides rich and dense supervision, ensuring that every sample contributes to learning even when no rollout is fully correct.

## 3.2 INFORMATION GAIN-BASED TURN-LEVEL REWARD

**Information Gain Reward.** We view agent–environment interaction as a process of *incrementally acquiring information about the ground truth*. To capture this intuition, we propose an intrinsic *information gain-based reward*. At each turn, we evaluate the model's ground-truth confidence and define the reward as the difference between consecutive interaction histories. We call this the *information gain reward*, as it measures the *ground-truth confidence gain* induced by the current turn. In practice, to ensure numerical stability and comparability across answers of different lengths, we quantify this confidence using the length-normalized log-probability of the ground-truth answer.

For rollout $o_i$, let $C_{i,t}$ denote the complete interaction history after turn $t$: $C_{i,0}$ is the original query, and for $1 \leq t < T_i$, $C_{i,t}$ additionally includes generated tokens and tool responses up to turn $t$.

To compute the ground-truth confidence at turn $t$, we append the ground-truth answer to the current interaction history $C_{i,t}$ in the same schema as a predicted answer, e.g., `<think>Now there is enough information to answer</think><answer>Ground Truth a</answer>`, and compute the teacher-forced log-probability of the ground-truth answer tokens under the current policy. Formally, let $a = (a_1, \ldots, a_L)$ denote the ground-truth answer tokens. The length-normalized ground-truth confidence score is computed as

$$s_{i,t} = \frac{1}{L} \sum_{j=1}^{L} \log \pi_\theta \left( a_j \mid C_{i,t}, a_{<j} \right), \qquad 0 \leq t < T_i. \tag{3}$$

For notational simplicity, Eq. 3 omits the fixed wrapper tokens that precede and surround the answer.

Then, for turn $t$ in rollout $o_i$, the information gain reward is defined as the increment of this score between two consecutive interaction histories, with $\mathrm{sg}(\cdot)$ denoting the stop-gradient operator:

$$r_{i,t}^{\mathrm{IG}} \triangleq \mathrm{sg}(s_{i,t} - s_{i,t-1}), \qquad 1 \leq t < T_i. \tag{4}$$

This turn-level reward offers three key properties: (1) *ground-truth awareness*: It increases when the turn-action raises ground-truth confidence, and decreases otherwise. Crucially, this objective derivation minimizes the potential bias inherent in external reward models or manual process labeling. (2) *Dense supervision*: It provides turn-level dense signals that alleviate advantage collapse, enable fine-grained credit assignment, and improve sample efficiency. (3) *Computational efficiency*: Unlike other process reward designs, especially Monte Carlo estimation, it incurs negligible overhead, an advantage further amplified by the vectorized implementation presented below.

**Efficient Vectorized Implementation.** Although the information gain reward already enjoys inherent efficiency over existing process reward methods, we seek to further optimize its computation. A naive implementation computes the ground-truth confidence score separately for each prefix $C_{i,t}$, requiring multiple forward passes with approximate complexity $\sum_{t=0}^{T_i-1} L_{i,t}^2$, where $L_{i,t}$ is the sequence length of prefix $C_{i,t}$. To optimize this, we propose a vectorized implementation that appends $T_i$ copies of the formatted ground-truth target to the end of the trajectory, corresponding to prefixes $C_{i,0}, C_{i,1}, \ldots, C_{i,T_i-1}$. By

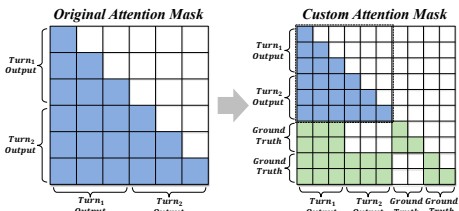

Figure 3: The custom attention mask for vectorized implementation. Shaded cells denote allowed attention. Prompt and answer tokens are omitted for clarity.

applying a custom attention mask (Figure 3) that restricts each copy's visibility to its corresponding prefix and its own previous ground truth tokens, we compute the log probabilities for all required turn prefixes simultaneously in a single forward pass, while ensuring mathematical equivalence to Eq. 3. The attention mask enforces the desired conditioning structure, while the appended target

length is orders of magnitude smaller than the full reasoning trajectory. Therefore, the overhead introduced by these appended copies is negligible in practice. This implementation reduces repeated forward passes and further enhances GPU utilization by reducing synchronization overhead.

### 3.3 POLICY OPTIMIZATION WITH TURN-LEVEL DISCOUNTED RETURN

**Turn-Level Discounted Return.** Given a group of rollouts $\{o_i\}_{i=1}^G$, each rollout $o_i$ yields a sequence of information gain rewards $\{r_{i,t}^{\mathrm{IG}}\}_{t=1}^{T_i-1}$ for intermediate turns and an outcome reward $r_i^{\mathrm{O}}$ for the final answer turn. Following GRPO (Shao et al., 2024), we stabilize training and capture the relative magnitude of rewards by performing group-wise $z$-normalization on the information gain rewards and outcome rewards separately.

Let $\mu_{\mathrm{IG}}$ and $\sigma_{\mathrm{IG}}$ denote the mean and standard deviation of all information gain rewards $\{r_{i,t}^{\mathrm{IG}} : 1 \leq i \leq G,\ 1 \leq t < T_i\}$ within the group. Similarly, let $\mu_{\mathrm{O}}$ and $\sigma_{\mathrm{O}}$ denote the mean and standard deviation of outcome rewards $\{r_i^{\mathrm{O}}\}_{i=1}^G$. The normalized reward is defined as

$$\tilde{r}_{i,t} = \begin{cases} \dfrac{r_{i,t}^{\mathrm{IG}} - \mu_{\mathrm{IG}}}{\sigma_{\mathrm{IG}}}, & 1 \leq t < T_i, \\[3ex] \dfrac{r_i^{\mathrm{O}} - \mu_{\mathrm{O}}}{\sigma_{\mathrm{O}}}, & t = T_i. \end{cases} \tag{5}$$

This separate normalization balances the scales of information-gain and outcome rewards, preventing either reward type from dominating the other.

While $\tilde{r}_{i,t}$ captures the relative quality of each turn, it only reflects immediate effects and ignores the impact of current decisions on future turns. To incorporate such long-horizon dependencies, we compute a turn-level discounted return $\tilde{R}_{i,t}$ to reflect the cumulative impact of future rewards:

$$\tilde{R}_{i,t} = \sum_{k=t}^{T_i} \gamma^{k-t} \tilde{r}_{i,k}, \qquad 1 \leq t \leq T_i, \tag{6}$$

where $\gamma \in (0, 1]$ is the discount factor. To assign turn-level returns to generated tokens, for each generated token $u_{i,m}$, let $\kappa_i(m) \in \{1, \dots, T_i\}$ denote the turn index in which $u_{i,m}$ is generated. During optimization, $\tilde{R}_{i,\kappa_i(m)}$ is assigned to token $u_{i,m}$. This yields a dense and future-aware supervision signal for policy learning.

**Policy Optimization.** With the turn-level discounted return $\tilde{R}_{i,t}$ defined above, we optimize the agent policy using a clipped surrogate objective with KL regularization, following the same structure as GRPO but with a finer-grained credit assignment. Formally, the IGPO objective is

$$\mathcal{J}_{\mathrm{IGPO}}(\theta) = \mathbb{E}_{(q,a) \sim \mathcal{D},\ \{o_i\}_{i=1}^G \sim \pi_{\theta_{\mathrm{old}}}^{\mathcal{E}}(\cdot|q)} \left[ \frac{1}{G} \sum_{i=1}^G \frac{1}{M_i} \sum_{m=1}^{M_i} \left( \min\left( \rho_{i,m}(\theta) \tilde{R}_{i,\kappa_i(m)}, \right. \right. \right.$$
$$\left. \left. \left. \mathrm{clip}(\rho_{i,m}(\theta),\ 1-\epsilon,\ 1+\epsilon)\, \tilde{R}_{i,\kappa_i(m)} \right) - \beta\, \mathbb{D}_{\mathrm{KL}}(\pi_\theta \,\|\, \pi_{\mathrm{ref}}) \right) \right], \tag{7}$$

where $\epsilon$ is the clipping threshold, and $\beta$ controls the token-level KL penalty relative to the reference policy $\pi_{\mathrm{ref}}$. Only model-generated decision tokens receive gradient updates, while raw tool responses are used as conditioning context and masked out from the loss.

To further substantiate the simplicity and implementability of the proposed IGPO, we provide an algorithmic flow comparison between IGPO and GRPO in Appendix G.

## 4 EXPERIMENTS

### 4.1 EXPERIMENTAL SETUP

**Datasets & Metrics.** To evaluate the effectiveness of our proposed IGPO, we conduct experiments on both in-domain (ID) and out-of-domain (OOD) QA benchmarks in an agentic search setting.

Table 1: Main results of IGPO compared with different agentic RL baselines across seven datasets.

| Method | In-domain | | | | Out-of-domain | | | |
|---|---|---|---|---|---|---|---|---|
| | NQ | TQ | HotpotQA | 2Wiki | Musique | Bamboogle | PopQA | Avg. |
| **Prompt-based** | | | | | | | | |
| CoT | 19.8 | 45.6 | 24.4 | 26.4 | 8.5 | 22.1 | 17.0 | 23.4 |
| CoT+RAG | 42.0 | 68.9 | 37.1 | 24.4 | 10.0 | 25.4 | 46.9 | 36.4 |
| Search-o1 | 32.4 | 58.9 | 33.0 | 30.9 | 14.7 | 46.6 | 38.3 | 36.4 |
| **Outcome-reward RL-based** | | | | | | | | |
| Search-r1-base | 45.4 | 71.9 | 55.9 | 44.6 | 26.7 | 56.5 | 43.2 | 49.2 |
| Search-r1-instruct | 33.1 | 44.7 | 45.7 | 43.4 | 26.5 | 45.0 | 43.0 | 40.2 |
| R1-searcher | 35.4 | 73.1 | 44.8 | 59.4 | 22.8 | 64.8 | 42.7 | 49.0 |
| DeepResearcher | 39.6 | 78.4 | 52.8 | 59.7 | 27.1 | 71.0 | 48.5 | 53.9 |
| **Step-reward RL-based** | | | | | | | | |
| StepSearch-base | - | - | 49.3 | 45.0 | 32.4 | 57.3 | - | 46.0 |
| StepSearch-instruct | - | - | 50.2 | 43.1 | 31.2 | 53.4 | - | 44.5 |
| ReasoningRAG | - | - | 48.9 | 50.4 | 20.6 | 45.5 | 46.2 | 42.3 |
| GiGPO | **46.4** | 64.7 | 41.6 | 43.6 | 18.9 | 68.9 | 46.1 | 47.2 |
| **IGPO** | **46.4** | **80.6** | **59.0** | **72.1** | **32.7** | **77.0** | **53.8** | **60.2** |

Following previous work (Zheng et al., 2025b; Deng et al., 2025), the ID setting includes four widely used datasets: NQ (Kwiatkowski et al., 2019), TQ (Joshi et al., 2017), HotpotQA (Yang et al., 2018), and 2Wiki (Ho et al., 2020), while the OOD setting includes three datasets: MusiQue (Trivedi et al., 2022), Bamboogle (Press et al., 2023), and PopQA (Mallen et al., 2022). We report word-level F1 as the evaluation metric, computed between predicted and reference answers as in Eq. 2.

**Baselines.** To evaluate IGPO on agentic search tasks, we compare it against strong baselines: (1) Prompt-based methods: CoT (Wei et al., 2022), CoT+RAG (Gao et al., 2023), and Search-o1 (Li et al., 2025b), which represent the baseline performance of LLMs without further training on search agent tasks. (2) Outcome-reward RL methods: Search-r1 (Jin et al.), R1-searcher (Song et al., 2025a), and DeepResearcher (Zheng et al., 2025b), the representative search agents with outcome reward-based RL, yielding marked performance gains. (3) Step-reward RL methods: StepSearch (Wang et al., 2025), ReasoningRAG (Zhang et al., 2026b), and GiGPO (Feng et al., 2026), which are the latest approaches exploring step-reward RL in search-agent settings.

To further validate IGPO's effectiveness, we also compare it against the following commonly used RL algorithms under the same configuration: PPO (Schulman et al., 2017), a widely used actor-critic algorithm that requires an additional value model, and critic-free methods Reinforce++ (Hu et al., 2025), RLOO (Kool et al., 2019; Ahmadian et al., 2024), GRPO (Shao et al., 2024), and GSPO(Zheng et al., 2025a) which perform advantage estimation over trajectory groups or batches.

**Implementation Details.** We use Qwen2.5-7B-Instruct as the main backbone and Qwen2.5-3B-Instruct for ablations (Yang et al., 2024). The training is conducted using the verl (Sheng et al., 2025) framework. The discount factor $\gamma$ is set to 1.0 with no future tuning. At each training step, we sample 32 prompts and generate 16 rollouts for each prompt. The maximum dialogue turns are set to 10. For the environment, we use the Google Search API as our tool. The settings of our experiments are consistent with DeepResearcher (Zheng et al., 2025b). For the other baselines in Table 1, we directly copy their reported results. For the controlled RL comparison in Table 2, all RL training methods, including IGPO and the baselines, use exactly the same hyperparameter configurations. Prompt templates are shown in Appendix H; additional details are in Appendix C.

### 4.2 OVERALL PERFORMANCE

The overall performance comparison between IGPO and the baseline methods is presented in Table 1 and Table 2. Based on these results, we can draw the following key observations:

**Training-based methods consistently outperform prompt-based baselines.** As shown in Table 1, RL-based methods consistently surpass prompt-based approaches, confirming the importance of explicit policy optimization for LLM-based search agents.

Table 2: Main results of IGPO compared with different RL baselines across seven datasets.

| | In-domain | | | | Out-of-domain | | | |
|---|---|---|---|---|---|---|---|---|
| Method | NQ | TQ | HotpotQA | 2Wiki | Musique | Bamboogle | PopQA | Avg. |
| RLOO | 40.7 | 72.5 | 49.6 | 55.0 | 24.8 | 62.2 | 43.1 | 49.7 |
| PPO | 38.7 | 75.4 | 48.6 | 59.7 | 26.2 | 63.4 | 48.7 | 51.5 |
| GRPO | 40.3 | 77.0 | 48.9 | 57.7 | 25.0 | 65.1 | 49.6 | 51.9 |
| Reinforce++ | 34.3 | 67.5 | 45.9 | 54.5 | 23.7 | 61.2 | 44.3 | 47.3 |
| GSPO | 41.5 | 77.7 | 46.3 | 60.1 | 25.4 | 67.6 | 45.4 | 52.0 |
| **IGPO** | **46.4** | **80.6** | **59.0** | **72.1** | **32.7** | **77.0** | **53.8** | **60.2** |

**Existing step-reward methods yield competitive but unstable gains over outcome-reward methods.** While step-reward baselines occasionally surpass outcome-reward ones on specific datasets (e.g., StepSearch on Musique), their overall performance still lags behind strong outcome-reward methods such as DeepResearcher. This suggests that existing step-reward designs, though providing intermediate guidance, often suffer from noisy or weak signals that limit generalizability.

**IGPO achieves the best overall performance across both in-domain and out-of-domain datasets.** Our IGPO outperforms all baselines, with an average score of 60.2, a clear margin over the best method (+6.3 over DeepResearcher). This improvement is attributed to IGPO's information gain-based reward design, which assigns intrinsic, ground-truth-aware credit at every turn while preserving the outcome reward. By providing fine-grained credit assignment and improving sample efficiency, IGPO delivers robust gains across both in-domain and out-of-domain datasets.

**IGPO consistently outperforms other RL algorithms.** Beyond task-specific baselines, Table 2 shows that IGPO also achieves the highest overall score among standard RL methods, surpassing RLOO, PPO, GRPO, Reinforce++, and GSPO. Unlike these methods, which rely solely on sparse outcome rewards, IGPO incorporates turn-level discounted returns to provide denser and more stable supervision, leading to stronger generalization and more efficient training.

## 4.3 ABLATION STUDY

We further conduct ablation experiments to assess the contribution of different reward components. As shown in Table 3, we observe:

**First, using only information gain (IG) turn-level reward or only outcome reward (F1) yields clearly inferior results compared to the full combination.** This highlights the complementary roles of turn-level and outcome-level supervision: the outcome reward enforces alignment with the final task objective but suffers from severe sparsity, whereas the information gain reward offers dense and stable guidance for intermediate steps.

**Second, IGPO with only IG achieves performance comparable to or even exceeding that of standard GRPO (i.e., IGPO w/ F1).** This suggests that IGPO's information gain reward does not suffer from severe reward hacking in our setting. Usually, without outcome supervision, unstable reward designs would quickly collapse. In contrast, our IGPO remains robust because its turn-level signals are intrinsically defined and grounded in the ground truth.

**Third, the improvements are particularly pronounced on the smaller 3B model.** Compared to standard GRPO, IGPO improves the 3B model by +16.6 points ($32.3 \rightarrow 48.9$) and the 7B model by +8.3 points ($51.9 \rightarrow 60.2$). This larger benefit on the 3B model arises because advantage collapse is more severe for weaker models that struggle to directly produce correct answers (Figure 1), making them especially reliant on dense reward signals. In such cases, the information gain reward helps prune noisy reasoning paths and reinforce rollouts that progressively approach the ground truth.

**Finally, IGPO demonstrates faster and more stable learning dynamics.** As shown in Figure 4, IGPO steadily outperforms its two ablated variants across all seven datasets. The curves highlight two advantages: (i) IGPO converges to higher F1 scores, confirming the benefit of combining intrinsic turn-level and outcome rewards, and (ii) IGPO maintains stable improvements over training, indicating robustness against reward sparsity and noisy supervision. These results further validate that IGPO provides dense, reliable signals, improving both training efficiency and final performance.

Table 3: Ablation results of IGPO on Qwen2.5-3B/7B-Instruct with different reward designs. IGPO (w/ F1) corresponds to using only outcome rewards, reducing to standard GRPO.

| Method | In-domain | | | | Out-of-domain | | | |
| --- | --- | --- | --- | --- | --- | --- | --- | --- |
| | NQ | TQ | HotpotQA | 2Wiki | Musique | Bamboogle | PopQA | Avg. |
| **Qwen2.5-3B-Instruct** | | | | | | | | |
| IGPO (w/ F1) | 31.0 | 55.6 | 27.5 | 29.4 | 12.1 | 35.7 | 34.9 | 32.3 |
| IGPO (w/ IG) | 29.6 | 54.1 | 28.1 | 37.5 | 17.6 | 43.8 | 31.7 | 34.6 |
| IGPO (w/ F1+IG) | **41.9** | **69.2** | **47.8** | **51.4** | **24.8** | **58.4** | **49.0** | **48.9** |
| **Qwen2.5-7B-Instruct** | | | | | | | | |
| IGPO (w/ F1) | 40.3 | 77.0 | 48.9 | 57.7 | 25.0 | 65.1 | 49.6 | 51.9 |
| IGPO (w/ IG) | 37.3 | 75.2 | 52.1 | 63.3 | 28.9 | 69.8 | 47.8 | 53.5 |
| IGPO (w/ F1+IG) | **46.4** | **80.6** | **59.0** | **72.1** | **32.7** | **77.0** | **53.8** | **60.2** |

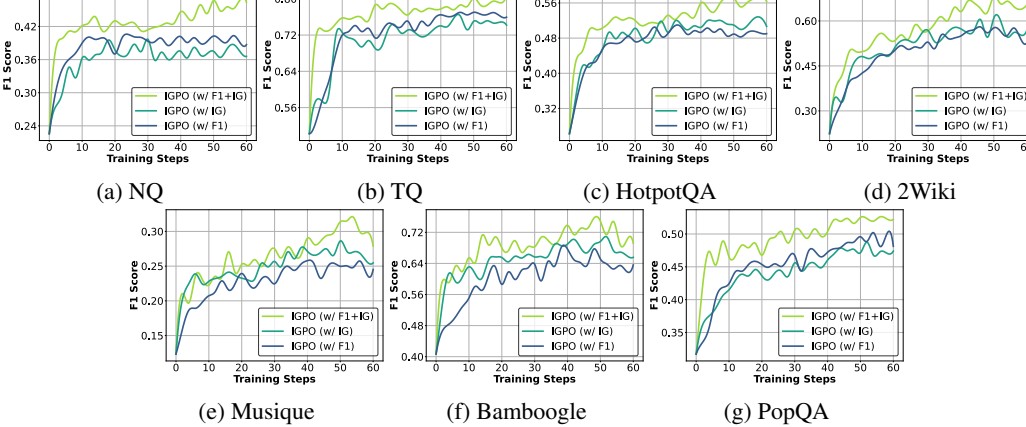

(a) NQ      (b) TQ      (c) HotpotQA      (d) 2Wiki

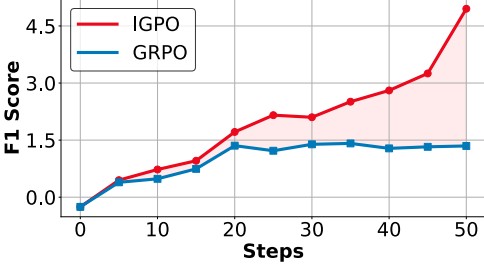
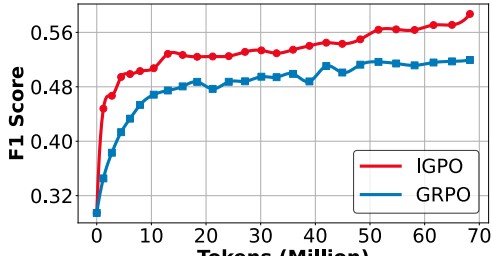

(e) Musique      (f) Bamboogle      (g) PopQA

Figure 4: Training curves on Qwen2.5-7B-Instruct with different reward designs.

In addition to the reward ablation, we compare different information gain bases (probability vs. log-probability) and normalization strategies (joint vs. separate) in Appendix D.1.

## 4.4 IN-DEPTH ANALYSIS

**Ground-truth Confidence Gain.** To better understand how IGPO improves training dynamics, we measure the change in ground-truth confidence from the initial query (Turn 0) to the last non-answer turn $(T-1)$. As shown in Figure 5, IGPO consistently achieves a larger confidence increase than GRPO throughout training. This indicates that the information gain reward effectively encourages intermediate steps to move the policy closer to the ground-truth answer distribution. In contrast, outcome-based supervision in GRPO provides no guidance for intermediate turns, resulting in weaker confidence increase. These results highlight that IGPO's turn-level supervision translates into more confident and grounded reasoning trajectories.

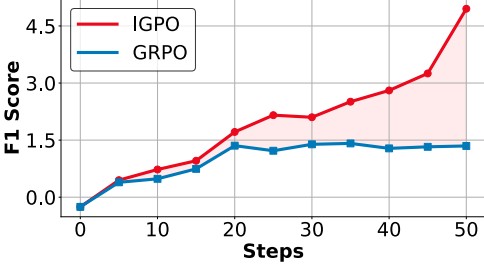

Figure 5: Mean ground-truth confidence gain before the final answer turn during training.

Figure 6: Token efficiency measured by average performance per gradient-update token.

**Token Efficiency.** We further compare IGPO and GRPO in terms of token efficiency, i.e., the performance improvement per token used for gradient updates. As shown in Figure 6, performance

increases more rapidly under IGPO, and the gap over GRPO widens during training. In other words, IGPO achieves stronger performance with fewer tokens, indicating that its turn-level rewards deliver denser and more informative gradients than outcome-only supervision. This finding is consistent with Figure 4, where IGPO converges faster while maintaining a stable performance lead.

Additional analyses are provided in Appendix D, including spurious-correlation analysis in Appendix D.4, failure modes in Appendix E, and case studies in Appendix I. Beyond empirical results, our theoretical analysis in Appendix A provides an intuitive perspective on how turn-level information gain may help constrain error accumulation in multi-turn scenarios.

### 4.5 COMPUTATIONAL BUDGET

We compare the runtime of IGPO and GRPO to quantify the overhead introduced by information-gain reward computation. As shown in Figure 7, IGPO incurs only 0.0227s additional time per step, corresponding to less than 0.4% overhead in the information-gain computation stage and less than 0.02% end-to-end training overhead. This confirms that IGPO provides fine-grained credit assignment with negligible computational cost. A detailed FLOPs analysis is provided in Appendix F.

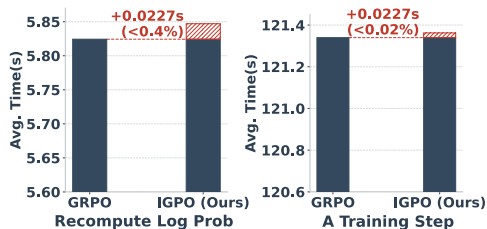

Figure 7: IGPO adds negligible overhead: 0.0227s per step and $< 0.02\%$ end-to-end.

## 5 RELATED WORK

The recent success of RL in large reasoning models (Chen et al., 2025), such as OpenAI o1 (Jaech et al., 2024) and DeepSeek R1 (Guo et al., 2025), has established RL as an effective paradigm for enhancing LLM-based agents. A growing body of work has explored RL algorithms including PPO (Schulman et al., 2017), Reinforce++ (Hu et al., 2025), GRPO (Shao et al., 2024), RLOO (Kool et al., 2019; Ahmadian et al., 2024), DAPO (Yu et al., 2026), and GSPO (Zheng et al., 2025a), which have been effective in improving LLM-based agent capabilities (Li et al., 2025a).

Building on these advances, another line of work applies RL to search-based agents (Deng et al., 2025; Dai et al., 2026; 2025). DeepRetrieval (Jiang et al.) uses retrieval-oriented rewards for end-to-end optimization, while Search-R1 (Jin et al.), DeepResearcher (Zheng et al., 2025b), Re-Search (Chen et al.), R1-Searcher (Song et al., 2025a), and R1-Searcher++ (Song et al., 2025b) extend RL to multi-turn reasoning and search interaction.

However, outcome-only rewards remain sparse in agentic scenarios, leading to unstable optimization and inefficient sample utilization. Recent studies therefore explore process-level rewards for intermediate actions. ReasonRAG (Zhang et al., 2026b) estimates step values with Monte Carlo Tree Search, StepSearch (Wang et al., 2025) uses retrieved-document similarity to ground-truth evidence, and GiGPO (Feng et al., 2026) estimates relative advantages from shared anchor states. While denser, these methods either rely on external oracle knowledge or suffer from limited stability and scalability, motivating more scalable and generalizable process-level rewards.

## 6 CONCLUSION, LIMITATIONS AND FUTURE WORK

In this work, we propose IGPO, a simple and effective RL framework for training LLM-based search agents. By measuring turn-level information gain toward the ground truth, IGPO provides dense, fine-grained supervision for intermediate steps. This design mitigates key limitations of outcome-only RL in agentic settings, including advantage collapse, weak credit assignment, and poor sample efficiency. Experiments across in-domain and out-of-domain benchmarks show that IGPO consistently outperforms strong baselines, achieving higher accuracy and training efficiency, especially for smaller models where sparse rewards are most problematic. Further analyses show that IGPO improves ground-truth confidence, enhances token efficiency, and adds negligible overhead.

However, IGPO still requires ground truth answers, limiting its use in open-ended settings. Future work will extend IGPO beyond search to broader agentic tasks without explicit supervision.

## ACKNOWLEDGMENTS

This work was supported by Ant Group Research Intern Program. We thank the Venus Team, Ant Group for their resource support and technical guidance.

## ETHICS STATEMENT

This work follows the ICLR Code of Ethics. No human or animal subjects were involved. All datasets (NQ, TQ, HotpotQA, 2Wiki, MuSiQue, Bamboogle, PopQA) were used according to their respective guidelines, with privacy fully respected. No personally identifiable information was included, and all procedures avoided potential privacy or security risks. Research was conducted with transparency and integrity.

## REPRODUCIBILITY STATEMENT

Every effort has been made to ensure the reproducibility of the results reported in this paper. All code and experimental resources are publicly available in the project repository to facilitate replication and verification. The experimental setup—including training procedures, model configurations, and hardware specifications—is detailed in the appendix to support accurate reproduction of the experiments. These measures are intended to enable other researchers to reproduce the work and contribute to further advancements in the field.

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

# A    THEORETICAL ANALYSIS

The theoretical analysis here provides an intuitive perspective on the efficacy of our proposed method by relating turn-level process rewards to the accumulation of snowball errors in multi-turn agents. Specifically, the analysis suggests a link between maximizing the process reward used by IGPO and reducing an upper bound on the undesirable accumulation of snowball errors during the reasoning process. This perspective helps explain why IGPO's dense, turn-level signals can lead to more confident and successful reasoning trajectories.

**Notations**. Let $E_{\text{final}}$ be the event that the agent's generated final answer does not match the ground truth answer. Its probability is denoted by $\mathbb{P}(E_{\text{final}})$, i.e., the error rate. For each turn $t$ , denote the observed response [think], [tool call] as $\mathcal{R}_t$. We also posit that there is an unobservable, abstract thinking step $\mathcal{I}_t$ that underlies the generation of $\mathcal{R}_t$. Let $R_{\text{process}}^{(t)}$ be the process reward, which is a dense reward signal received at each turn of the interaction. It is defined as the information gain about the ground truth answer, instantiated as the ground-truth confidence gain from the previous state to the current state. Then, the total process reward $R_{\text{total}} = \sum_{t=1}^{T-1} \mathbb{E}[R_{\text{process}}^{(t)}]$ is the cumulative sum of all process rewards over a complete trajectory or episode. The expectation is taken over the thinking step and observed response. The training objective of the policy is to maximize this total reward.

**Definition A.1** (Snowball Error in Multi-turn Agentic RL). *Consistent with Gan et al. (2025), we define the information loss at turn $t$ as the conditional entropy $\text{Ent}(\mathcal{I}_t|\mathcal{R}_t)$. Consider the non-trivial case where $|\text{Ent}(\mathcal{I}_t|\mathcal{R}_t)|$ is bounded. The cumulative snowball error up to turn $T$ is the sum of these losses:*

$$\text{Ent}_{<T}(\mathcal{I}|\mathcal{R}) \triangleq \sum_{t=1}^{T-1} \text{Ent}(\mathcal{I}_t|\mathcal{R}_t) \tag{8}$$

This quantity measures the aggregate uncertainty and ambiguity accumulated throughout the reasoning trajectory before the final answer is produced.

Next, we connect the cumulative snowball error to the agent's final performance. It indicates the fundamental limitation of multi-turn agentic RL pipeline caused by snowball error.

**Lemma A.2** (Lower bound of error rate). *The probability of a final answer error, $\mathbb{P}(E_{final})$, is lower-bounded by the cumulative snowball error accumulated during the reasoning process:*

$$\mathbb{P}(E_{final}) \geq \Omega\left(\frac{\text{Ent}_{<T}(\mathcal{I}|\mathcal{R})}{T-1}\right) - C_{const}. \tag{9}$$

*where $C_{const}$ is a small positive constant.*

*Proof Sketch.* This result is strongly motivated by Theorem 3.3 from Gan et al. (2025). We treat the generation of the final answer at turn $T$ as the final step of a multi-step reasoning process. The quality of this final step is conditioned on the information accumulated over the previous $T-1$ turns. The theorem from (Gan et al., 2025) states that the error probability of any step is lower-bounded by the average snowball error accumulated up to that point. Applying this principle to the final step ($t = T$) yields the stated result. $\square$

**Assumption A.3** (Monotonic Reward-Information Loss Link). *The expected process reward at any turn $t$, $\mathbb{E}[R_{process}^{(t)}]$, is monotonically non-increasing with respect to the information loss at that turn, $\text{Ent}(\mathcal{I}_t|\mathcal{R}_t)$. We assume there exists a bounded and monotonically non-increasing function $f : \mathbb{R}_+ \to \mathbb{R}$ such that:*

$$\mathbb{E}[R_{process}^{(t)}|\mathcal{I}_t, \mathcal{R}_t] \leq f\left(\text{Ent}(\mathcal{I}_t|\mathcal{R}_t)\right). \tag{10}$$

*Furthermore, we assume that there exist constants $C_{\max} > 0$ and $\beta_{\text{info}} > 0$ such that*

$$f(x) \leq C_{\max} - \beta_{\text{info}}x \tag{11}$$

*for the bounded range of information loss considered here.*

**Remark.** *This assumption captures the intuition that turns with lower information loss tend to yield higher expected process rewards.*

This assumption leads to the following result, demonstrating that optimizing for process rewards implicitly constrains the accumulation of snowball errors. We first formalize the intuition that a clearer reasoning step (lower information loss) is a prerequisite for a high-quality query, which in turn yields a higher expected process reward.

**Theorem A.4** (Process Reward as a Bound on Snowball Error). *Under Assumption A.3, the expected cumulative snowball error is upper bounded by*

$$\mathbb{E}[\text{Ent}_{<T}(\mathcal{I}|\mathcal{R})] \leq \frac{(T-1)C_{\max} - R_{total}}{\beta_{\text{info}}}, \tag{12}$$

*where $C_{\max}$ and $\beta_{\text{info}}$ are positive constants.*

Theorem A.4 suggests that increasing the process reward tightens an upper bound on the cumulative snowball error. Together with Lemma A.2, this provides an intuitive explanation for why dense turn-level rewards can improve multi-turn agent training. The logical chain is as follows:

- **Increasing the process reward** is associated with a **smaller upper bound on the cumulative snowball error** (Theorem A.4).

- A smaller cumulative snowball error reduces the lower-bound term associated with the final error rate, offering an intuitive explanation for improved task success (Lemma A.2).

In conclusion, this analysis suggests that the turn-level process reward is more than an engineering heuristic: it provides a principled way to mitigate error accumulation in multi-step reasoning. By providing a dense, immediate signal for reasoning clarity, it transforms the intractable problem of sparse-reward, long-horizon exploration into a series of manageable, short-horizon sub-problems, each aimed at maximizing immediate information gain. This explains the significant gains in training efficiency and final performance observed in our experiments.

## B  PROOF FOR THEORETICAL ANALYSIS

### B.1  PROOF OF LEMMA A.2

*Proof.* We achieve this by applying Theorem 3.3 from Gan et al. (2025) to the final decision-making step of the agent. In particular,

$$\mathbb{P}(E_{\text{final}}) \geq \frac{\frac{\text{Ent}_{<T}(\mathcal{I}|\mathcal{R})}{T-1} - C_1}{\log(|\mathcal{A}_{\text{final}}| - 1)}, \tag{13}$$

where $|\mathcal{A}_{\text{final}}|$ is the cardinality of the final answer space and $C_1$ is a small positive constant analogous to $\text{Ent}_b(e_t)$ in Gan et al. (2025). Since $\log(|\mathcal{A}_{\text{final}}|-1)$ and $C_1$ are constant, $\frac{\frac{\text{Ent}_{<T}(\mathcal{I}|\mathcal{R})}{T-1} - C_1}{\log(|\mathcal{A}_{\text{final}}|-1)}$ simplifies to a form that is asymptotically dominated by the variable term. Therefore, the right-hand side of the inequality can be expressed in terms of the lower bound symbol $\Omega$ as $\Omega\left(\frac{\text{Ent}_{<T}(\mathcal{I}|\mathcal{R})}{T-1}\right) - C_{\text{const}}$, which completes the proof. $\square$

### B.2  PROOF OF THEOREM A.4

*Proof.* According to the nature of $f$ and the fact that there exist constants $C_{\max}$ and $\beta_{\text{info}}$ such that for all non-negative bounded $x$, there holds $f(x) \leq C_{\max} - \beta_{\text{info}}x$. Therefore, by taking the expectation over Assumption A.3 and summing across all turns from $t = 1$ to $T - 1$, we have

$$R_{\text{total}} = \sum_{t=1}^{T-1} \mathbb{E}[R_{\text{process}}^{(t)}] \leq \sum_{t=1}^{T-1} \mathbb{E}[f(\text{Ent}(\mathcal{I}_t|\mathcal{R}_t))]$$

$$\leq \sum_{t=1}^{T-1} \mathbb{E}[C_{\max} - \beta_{\text{info}} \cdot \text{Ent}(\mathcal{I}_t|\mathcal{R}_t)]$$

$$= (T-1)C_{\max} - \beta_{\text{info}} \sum_{t=1}^{T-1} \mathbb{E}[\text{Ent}(\mathcal{I}_t|\mathcal{R}_t)]$$

$$= (T-1)C_{\max} - \beta_{\text{info}} \mathbb{E}[\text{Ent}_{<T}(\mathcal{I}|\mathcal{R})].$$

Rearranging terms yields the final result. □

## C MORE IMPLEMENTATION DETAILS

All our training experiments are conducted on 8 × NVIDIA A100-80G GPUs. The detailed hyperparameter settings are provided in Table 4. Unless otherwise specified, all experiments are based on this configuration.

Table 4: Training hyperparameters.

| Training hyperparameters | Value |
|---|---|
| Training Batch Size | 32 |
| Mini-Batch Size | 512 |
| Infer Tensor Model Parallel Size | 1 |
| Sequence Parallel Size | 4 |
| Max Prompt Length | 30767 |
| Max Response Length | 2000 |
| Actor Learning Rate | 1e-6 |
| Rollout Temperature | 1.0 |
| Rollout Group Size | 16 |
| Max Turn Call Turns | 10 |
| KL-Divergence loss coefficient | 0.001 |

Table 5: Performance comparison across different IG bases and normalization strategies. We compare Prob-based IG (Joint & Separate normalization) against LogProb-based IG (Separate normalization). The combination of LogProb and Separate normalization outperforms other settings.

| Method | In-domain | | | | Out-of-domain | | | |
|---|---|---|---|---|---|---|---|---|
| | NQ | TQ | HotpotQA | 2Wiki | Musique | Bamboogle | PopQA | Avg. |
| **Qwen2.5-3B-Instruct** | | | | | | | | |
| Prob+Joint | 40.5 | **69.4** | 46.8 | 48.2 | 23.1 | 57.9 | 47.4 | 47.6 |
| Prob+Separate | 41.2 | 68.9 | 47.2 | 49.5 | 23.5 | 57.7 | 48.3 | 48.0 |
| Logprob+Separate | **41.9** | 69.2 | **47.8** | **51.4** | **24.8** | **58.4** | **49.0** | **48.9** |
| **Qwen2.5-7B-Instruct** | | | | | | | | |
| Prob+Joint | **46.7** | 80.1 | 57.2 | 68.2 | 31.4 | 74.9 | 52.5 | 58.7 |
| Prob+Separate | 46.2 | 80.3 | 58.2 | 71.4 | 31.8 | 74.6 | 53.6 | 59.4 |
| Logprob+Separate | 46.4 | **80.6** | **59.0** | **72.1** | **32.7** | **77.0** | **53.8** | **60.2** |

## D MORE DISCUSSION AND EXPERIMENTAL ANALYSIS

### D.1 COMPARISON OF INFO. GAIN BASIS (PROB. VS. LOGPROB) AND NORMALIZATION STRATEGIES (JOINT VS. SEPARATE)

We investigate the impact of different information gain computation bases (probability vs. log-probability) and normalization strategies: joint (normalizing all rewards collectively) vs. separate (normalizing IG and outcome independently). Note that we exclude the Logprob+Joint combination due to the significant scale disparity between log-based IG and bounded outcome rewards, which renders joint normalization ineffective. As shown in Table 5, the combination of log-probability-based information gain and separate normalization (Logprob+Separate) emerges as the optimal strategy. Specifically, switching from joint to separate normalization (Prob+Joint → Prob+Separate) yields a clear gain (+0.7 on 7B Avg, +0.7 on 3B Avg), validating the necessity of decoupling the statistics of intermediate and final rewards. Replacing probability with log-probability (Prob+Separate → Logprob+Separate) provides an additional boost (+0.8 on 7B Avg, +0.9 on 3B Avg), demonstrating the numerical stability advantages of log-probability. These improvements are

consistent across both model scales and domain types, demonstrating the robustness of our design choices.

## D.2 COMPARISON WITH OTHER PROCESS-REWARD METHODS

In addition to its obvious performance advantages, we also conduct a deeper analysis of IGPO's superiority in terms of algorithmic characteristics compared to other process-reward-based agentic RL algorithms. We first introduce other existing process-reward-based agentic RL algorithms:

- **ReasoningRAG**. The main contribution of this work is the proposal of a step-level data labeling strategy based on MCTS. Subsequently, the DPO algorithm is used to optimize the agent's policy on the labeled step-level dataset. The main limitations of this method are: (1) the data labeling process relies on MCTS, which is inefficient, and when the number of samples is insufficient, it is difficult to accurately estimate the value of each step; (2) the off-policy optimization based on DPO is less effective than on-policy algorithms.

- **StepSearch**. StepSearch constructs turn-level supervision signals by pre-defining golden search keywords and golden tool responses, and adopts an on-policy optimization approach. Although it shifts from off-policy to on-policy, the annotation process is resource-intensive and prone to annotator bias (whether from humans or LLMs).

- **GiGPO**. GiGPO introduces a step-level grouping strategy based on anchor states and performs fine-grained advantage estimation within each step-level group. Although this provides a novel solution, it essentially still relies on the Monte Carlo assumption. When the number of anchor states is insufficient, it is often difficult to accurately estimate their value, which in turn leads to biased advantage estimation.

The proposed IGPO effectively addresses the aforementioned limitations. Starting from the on-policy GRPO setting (where rollout data are used for a single parameter update), it employs an information-gain–based incremental reward construction strategy that requires no annotation and does not rely on Monte Carlo. Moreover, the incorporation of ground-truth awareness substantially reduces bias. Table 6 provides a detailed comparison highlighting the advantages of IGPO over other algorithms.

Table 6: Comparison between various process reward methods.

| Algorithm | On-Policy | Explicit Labeling-Free | Monte Carlo–Free | Introduces No Bias |
|---|---|---|---|---|
| ReasoningRAG | No | Yes | No | Sample-size Dependent |
| StepSearch | Yes | No | Yes | No |
| GiGPO | Yes | Yes | No | Sample-size Dependent |
| IGPO | Yes | Yes | Yes | Yes |

## D.3 TIME BREAKDOWN OF EACH STAGE IN IGPO TRAINING (SAME AS GRPO)

Table 7: We have calculated the average time percentage spent on each phase of a training step for IGPO (same as GRPO). The majority of the time is spent on Sampling (Rollout), with Recompute Log-Prob accounting for less than 5% of the total duration. The time spent on the Return/Advantage Computation phase is much smaller than 1% and can be ignored.

| Phase | Sampling | Param Update | Recompute Log-Prob | Return/Advantage Comp. |
|---|---|---|---|---|
| Time Proportion | 82.6% | 12.6% | 4.8% | $\ll 1\%$ |

## D.4 SPURIOUS CORRELATIONS ANALYSIS

It is widely acknowledged that LLMs often exploit spurious correlations to solve problems—achieving correct answers through unfaithful or erroneous intermediate reasoning—rather than learning the genuine underlying reasoning process. This tendency significantly compromises

their out-of-distribution performance. To investigate whether the proposed IGPO mitigates or exacerbates such spurious correlations, we conduct the following experiment:

**Experiment Setup.**    We select the set of test samples correctly answered (F1=1.0) by both IGPO and GRPO agents.  We extract the corresponding outputs, remove the final answers, and retain only the intermediate reasoning traces. Subsequently, we employ a powerful teacher LLM (gemini-2.5-pro (Comanici et al., 2025)) to deduce the final answer based on these reasoning paths.  By comparing the accuracy of the answers inferred from the IGPO versus GRPO reasoning traces, we assess whether IGPO is more prone to yielding 'correct answers with incorrect or unfaithful reasoning' (i.e., spurious correlations).  The prompt template used for gemini-2.5-pro is illustrated in Figure 10.

**Result.**    As shown in Table 8, for both 3B and 7B models, the accuracy of the teacher model in inferring answers from IGPO-agent traces consistently outperforms that from GRPO-agent traces across all seven datasets, including both in-domain and out-of-domain tasks.  This indicates that the reasoning traces generated by the IGPO agent are more informative and of higher quality. It demonstrates that IGPO effectively mitigates spurious correlations through ground-truth-guided fine-grained credit assignment, further validating its generalization capabilities.

**Additional Evidence.**    Beyond the aforementioned experiment, we provide the following evidence to further support that IGPO mitigates spurious correlations:  (i) **Superior OOD Performance**. According to the results in Table 3, compared to GRPO, IGPO achieves an average performance gain of 12.6% (7B) and 42.8% (3B) on In-domain datasets (NQ, TQ, HotpotQA, and 2Wiki), whereas on Out-of-domain datasets (Musique, Bamboogle, PopQA), the average improvement increases to 13.7% (7B) and 55.2% (3B). The fact that performance gains in OOD settings exceed those in ID settings contradicts the pattern of spurious correlations, which typically favors ID performance at the expense of OOD generalization. (ii) **Exceptional Multi-hop Capabilities**. As indicated in Table 3, IGPO outperforms GRPO with an average improvement of 7.4% (7B) and 29.5% (3B) on single-hop tasks (NQ, TQ, PopQA), while on multi-hop tasks (HotpotQA, 2Wiki, Musique, Bamboogle), the average improvement reaches 17.8% (7B) and 68.1% (3B). The performance boost on multi-hop tasks is significantly greater than on single-hop tasks.  This is also inconsistent with spurious correlation patterns, which are prone to appearing in multi-hop scenarios and consequently causing greater detriment to performance. Therefore, the superior performance in both OOD and multi-hop scenarios serves as further evidence that IGPO effectively mitigates spurious correlations.

Table 8: Results of spurious correlations analysis. We select test samples where both IGPO and GRPO agents achieve correct answers (F1=1.0).  Using gemini-2.5-pro, we infer answers solely based on the reasoning traces from these samples to compare the informativeness of the traces generated by each method ("All" denotes the aggregated accuracy across all test samples). The results demonstrate that, compared to GRPO, IGPO yields reasoning traces that are more informative and of higher quality, further validating its generalization capabilities.

| Method | In-domain | | | | Out-of-domain | | | |
| --- | --- | --- | --- | --- | --- | --- | --- | --- |
| | NQ | TQ | HotpotQA | 2Wiki | Musique | Bamboogle | PopQA | All |
| **Qwen2.5-3B-Instruct** | | | | | | | | |
| GRPO | 93.1 | 89.0 | 90.1 | 94.7 | 84.3 | 89.7 | 93.5 | 91.2 |
| IGPO | **95.4** | **92.8** | **94.2** | **97.7** | **90.2** | **92.3** | **96.1** | **94.6** |
| **Qwen2.5-7B-Instruct** | | | | | | | | |
| GRPO | 86.5 | 88.8 | 85.8 | 92.0 | 76.1 | 91.2 | 95.1 | 89.0 |
| IGPO | **89.5** | **91.1** | **92.6** | **94.4** | **83.0** | **94.1** | **95.1** | **92.0** |

## E    FAILURE ANALYSIS

Despite IGPO's superior performance, to ensure a comprehensive analysis, we investigated its failure modes,  specifically examining instances where IGPO exhibits performance degradation (i.e., F1

scores lower than GRPO). As shown in Table 9, minor degradation is observed across datasets, with IGPO underperforming GRPO on approximately 3.6% of the test samples overall. While this degradation is marginal, it warrants in-depth analysis.

Algorithmically, IGPO extends GRPO by computing the ground-truth confidence gain between adjacent turns. This constructs a turn-level reward, providing stronger, denser, and ground-truth-aware supervision for fine-grained guidance. While GRPO relies solely on final outcome correctness, both methods depend on ground truth quality. Consequently, while IGPO amplifies the benefits of high-quality data, it inevitably exacerbates the impact of noise within the ground truth.

Through detailed data inspection, we identified a representative pattern of ground truth failure: ambiguous questions lacking specific conditions, leading to multiple valid answers. In such scenarios, IGPO is prone to degradation. If the model leverages reasoning and tool usage to increase confidence in a factually correct—but non-ground truth—answer, it incurs a penalty from the turn-level reward. This erroneously suppresses valid behaviors and impairs the model's reasoning capabilities. Figure 8 illustrates a real training instance from HotpotQA. The question, "Who is the author of Childhood?", lacks context (e.g., genre), allowing for multiple valid answers. When the model retrieved information regarding "Nathalie Sarraute" (a correct answer, though not the designated ground truth), it was penalized (Info Gain = -0.81). This constitutes a false suppression of correct reasoning and tool usage. While GRPO also struggles with such ambiguity, IGPO amplifies this noise, resulting in a more significant negative impact. We observed frequent occurrences of this pattern in the training set and identify it as the primary cause of IGPO's performance degradation.

It is important to note, however, that this degradation stems from data defects rather than algorithmic flaws. The ability of IGPO to maintain high performance despite these data imperfections (with a degradation rate of only 3.6%) effectively demonstrates its robustness. We look forward to exploring more complex failure modes in future work.

Table 9: We compared the F1 scores of IGPO and GRPO on the test set and analyzed the proportion of samples falling into three categories: IGPO > GRPO, IGPO = GRPO, and IGPO < GRPO. Overall, IGPO exhibits slight performance degradation across datasets, with approximately 3.6% of test samples showing lower performance compared to GRPO.

| Dataset | IGPO>GRPO | IGPO=GRPO | IGPO<GRPO |
|---|---|---|---|
| 2Wiki | 35.8% | 59.6% | 4.6% |
| Bamboogle | 47.2% | 49.6% | 3.2% |
| HotpotQA | 49.2% | 48.4% | 2.4% |
| Musique | 71.2% | 25.4% | 3.4% |
| NQ | 57.4% | 40.4% | 2.2% |
| PopQA | 42.8% | 53.4% | 3.8% |
| TQ | 33.6% | 61.2% | 5.2% |
| All | 48.3% | 48.1% | 3.6% |

## F    COMPUTATION BUDGET THEORETICAL ANALYSIS

This section provides a detailed analysis of the additional FLOPs introduced by IGPO. Since the extra computation introduced by IGPO occurs during the `recompute_log_prob` phase (a single forward propagation), we will analyze the FLOPs of a single forward propagation in the Transformer model to examine the additional computation introduced by IGPO.

### F.1    FLOPs CALCULATION

FLOPs, short for floating-point operations, is commonly used to measure computational complexity. We will focus on the FLOPs calculation in matrix multiplication. For matrices $A \in \mathbb{R}^{1 \times n}$ and $B \in \mathbb{R}^{n \times 1}$, computing $AB$ requires $n$ multiplication operations and $n$ addition operations, totaling $2n$ FLOPs. Therefore, the FLOPs required to compute $AB$ for matrices $A \in \mathbb{R}^{m \times n}$ and $B \in \mathbb{R}^{n \times p}$ is $2nmp$.

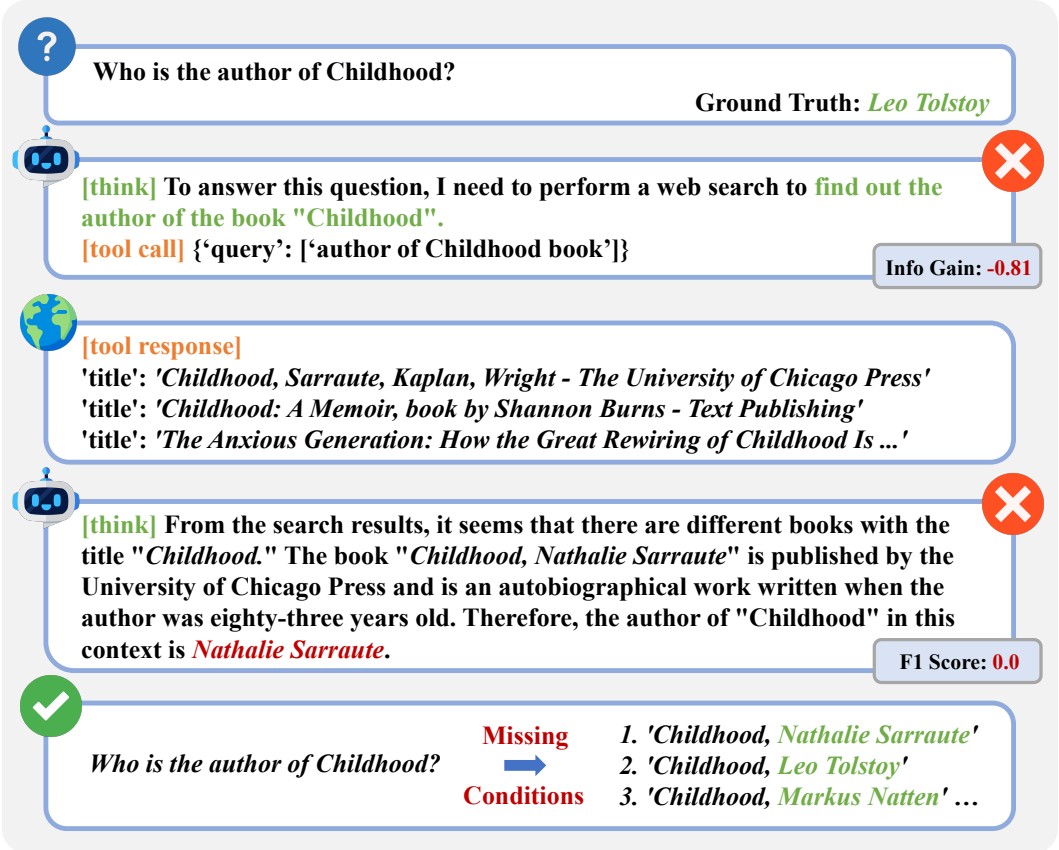

Figure 8: The query 'Who is the author of Childhood?' (a question from HotpotQA) is inherently ambiguous because multiple unrelated works share the same title—e.g., Childhood by Leo Tolstoy (fiction, ground truth), Nathalie Sarraute (autobiography, the model's answer), and Markus Natten (poem). Without specifying which literary work is intended, several factually correct answers exist. Consequently, when the model outputs a correct but non–ground truth author, it is penalized as 'wrong,' producing misleading negative rewards. Such mislabeled supervision degrades the effectiveness of IGPO by punishing valid reasoning aligned with alternative correct interpretations.

## F.2 SYMBOLS

Let $b$: batch_size, $s$: original trajectory sequence length before appending formatted ground-truth targets, $g$: token length of one formatted ground-truth target appended for teacher-forced confidence computation, $T$: number of formatted ground-truth target copies appended for computing turn-level confidence scores, $h$: hidden state dimension (assume the intermediate dimension $= 4h$), $n_{\text{head}}$: number of attention heads, $d_{\text{head}}$: dimension of each attention head, $l$: number of layers, $\mathcal{V}$: vocabulary size, $x$: the input data, $W_Q$: the query matrix, $W_K$: the key matrix, $W_V$: the value matrix, $W_o$: the output matrix of the attention module, $W_{up}$: the up-projection matrix in MLP module, $W_{down}$: the down-projection matrix in MLP module, $y_{attn}$: the output of attention module, $y_{mlp}$: the output of MLP module, $\sigma$: the activation function.

## F.3 ADDITIONAL FLOPS INTRODUCED BY IGPO COMPARED TO GRPO.

We first analyze the FLOPs of GRPO's `recompute_log_prob` phase, which refers to the FLOPs of a single forward propagation. The shape of the input data is $[b, s]$. We first analyze the self-attention module, whose computation process is as follows:

$$Q = xW_Q, \quad K = xW_K, \quad V = xW_V. \tag{14}$$

$$y_{\text{attn}} = \text{softmax}\left(\frac{QK^\top}{\sqrt{d_{\text{head}}}}\right) \cdot V \cdot W_o + x. \tag{15}$$

The input and output shapes of the matrix multiplication in Eq. 14 are: $[b, s, h] \times [h, h] \to [b, s, h]$. Based on the matrix multiplication rule above, the FLOPs for this process are:

$$\text{FLOPs}_{\text{attn1}} = 3 \times 2bsh^2 = 6bsh^2. \tag{16}$$

For $QK^\top$ in Eq. 15, the input and output shapes of the matrix multiplication are $[b, n_{\text{head}}, s, d_{\text{head}}] \times [b, n_{\text{head}}, d_{\text{head}}, s] \to [b, n_{\text{head}}, s, s]$, and the FLOPs for this process are $2bs^2h$.

For `attention_score`$\cdot V$ in Eq. 15, the input and output shapes are $[b, n_{\text{head}}, s, s] \times [b, n_{\text{head}}, s, d_{\text{head}}] \to [b, n_{\text{head}}, s, d_{\text{head}}]$, and the FLOPs for this process are $2bs^2h$.

For the linear mapping operation of multiplying by $W_o$ in Eq. 15, the input and output shapes of the matrix multiplication are: $[b, s, h] \times [h, h] \to [b, s, h]$, the FLOPs for this process are $2bsh^2$.

Therefore, the total FLOPs in Eq. 15 are:

$$\text{FLOPs}_{\text{attn2}} = 2bs^2h + 2bs^2h + 2bsh^2 = 4bs^2h + 2bsh^2. \tag{17}$$

Next, we analyze the FLOPs of the MLP module. For simplicity, we analyze a standard Transformer FFN with intermediate dimension $4h$; architecture-specific constants may vary, but the dependence on sequence length remains unchanged. The computation process of the MLP module can be expressed as follows:

$$y_{\text{mlp}} = \sigma(y_{\text{attn}}W_{\text{up}})W_{\text{down}} + y_{\text{attn}}. \tag{18}$$

For the up-projection operation in Eq. 18, the input and output shapes are $[b, s, h] \times [h, 4h] \to [b, s, 4h]$, and the FLOPs for this process are $8bsh^2$. For the down-projection operation, the input and output shapes are $[b, s, 4h] \times [4h, h] \to [b, s, h]$, and the FLOPs for this process are $8bsh^2$. Therefore, the total FLOPs of the MLP model are:

$$\text{FLOPs}_{\text{mlp}} = 2 \times 8bsh^2 = 16bsh^2. \tag{19}$$

At this point, we have calculated the FLOPs required for a Transformer block during the forward propagation process: $\text{FLOPs}_{\text{attn1}} + \text{FLOPs}_{\text{attn2}} + \text{FLOPs}_{\text{mlp}}$. Therefore, the FLOPs for the entire Transformer model are $l \times (\text{FLOPs}_{\text{attn1}} + \text{FLOPs}_{\text{attn2}} + \text{FLOPs}_{\text{mlp}})$.

In addition, due to the large size of the vocabulary $\mathcal{V}$, the computation involved in mapping the hidden state (dimension $= h$) to logits (dimension $= \mathcal{V}$) at the end is also significant and cannot be ignored. The input and output shapes of the matrix multiplication are: $[b, s, h] \times [h, \mathcal{V}]$, the FLOPs for this process are $\text{FLOPs}_{\text{logits}} = 2bsh\mathcal{V}$.

Therefore, the FLOPs of the GRPO's `recompute_log_prob` phase are:

$$\text{FLOPs}_{\text{GRPO}} = l \times (\text{FLOPs}_{\text{attn1}} + \text{FLOPs}_{\text{attn2}} + \text{FLOPs}_{\text{mlp}}) + \text{FLOPs}_{\text{logits}}$$
$$= 24lbsh^2 + 4lbs^2h + 2bsh\mathcal{V}. \tag{20}$$

For the sake of simplicity in the subsequent analysis, we let $\alpha_{\text{F}} = 4lbh$ and $\beta_{\text{F}} = 24lbh^2 + 2bh\mathcal{V}$. The FLOPs of GRPO can be expressed as:

$$\text{FLOPs}_{\text{GRPO}} = \alpha_{\text{F}}s^2 + \beta_{\text{F}}s. \tag{21}$$

Here, we use $\alpha_{\text{F}}$ and $\beta_{\text{F}}$ to avoid confusion with the KL coefficient $\beta$ in the policy optimization objective.

Since we integrate the computation of ground truth log-probabilities into the `recompute_log_prob` phase by appending formatted ground-truth targets and applying a custom attention mask, IGPO processes additional target tokens during this phase. Let $g$ denote the length of one formatted ground-truth target, and let $T$ denote the number of target copies appended for computing turn-level confidence scores. Thus, the total appended length is $Tg$, and the FLOPs for IGPO's `recompute_log_prob` phase are:

$$\text{FLOPs}_{\text{IGPO}} = \alpha_{\text{F}}(s + Tg)^2 + \beta_{\text{F}}(s + Tg) \tag{22}$$
$$\approx \alpha_{\text{F}}(s^2 + 2sTg) + \beta_{\text{F}}(s + Tg), \quad \text{since } Tg \ll s. \tag{23}$$

The approximation in Eq. 23 omits the second-order term $\alpha_{\text{F}}T^2g^2$, since the total appended length $Tg$ is much smaller than the full trajectory length $s$ during actual training.

The additional FLOPs introduced by IGPO compared to GRPO are:

$$\Delta\text{FLOPs} = \text{FLOPs}_{\text{IGPO}} - \text{FLOPs}_{\text{GRPO}}$$
$$\approx 2\alpha_{\text{F}}sTg + \beta_{\text{F}}Tg. \tag{24}$$

The proportion of additional FLOPs introduced compared to GRPO is:

$$\frac{\Delta\text{FLOPs}}{\text{FLOPs}_{\text{GRPO}}} = \frac{2\alpha_{\text{F}}sTg + \beta_{\text{F}}Tg}{\alpha_{\text{F}}s^2 + \beta_{\text{F}}s}$$
$$= \frac{Tg(2\alpha_{\text{F}}s + \beta_{\text{F}})}{s(\alpha_{\text{F}}s + \beta_{\text{F}})} \tag{25}$$
$$< \frac{2Tg}{s}.$$

In actual training scenarios, $\frac{2Tg}{s}$ remains very small because the formatted ground-truth target is short and the maximum number of turns is bounded. Therefore, the additional FLOPs introduced by IGPO's `recompute_log_prob` phase are negligible. Considering the real-time proportion of the `recompute_log_prob` phase in Table 7, IGPO introduces only negligible additional end-to-end training overhead compared to GRPO.

## G   COMPARISON BETWEEN GRPO AND IGPO

Algorithm 1 illustrates the algorithmic flow of GRPO (left) and IGPO (right). The key steps corresponding to each algorithm are highlighted with the same color font: yellow for reward calculation, green for return/advantage estimation, blue for credit assignment, and purple for policy optimization. In GRPO, the outcome-based advantage is assigned to all model-generated decision tokens in a rollout. In contrast, IGPO constructs turn-level information gain rewards, computes turn-level discounted returns, and assigns each generated token the return of its corresponding turn.

---

**Algorithm 1 GRPO** vs. **IGPO**

**GRPO**

**Require:** initial policy $\pi_{\theta_{\text{init}}}$; task prompts $\mathcal{D}$; tool environment $\mathcal{E}$; rollout group size $G$; number of training steps $S$; hyperparameters $\epsilon, \beta, \mu$

1: $\pi_\theta \leftarrow \pi_{\theta_{\text{init}}}$
2: **for** iteration $= 1, \ldots, I$ **do**
3:     $\pi_{\text{ref}} \leftarrow \pi_\theta$
4:     **for** step $= 1, \ldots, S$ **do**
5:        Sample a batch $\mathcal{D}_b$ from $\mathcal{D}$
6:        $\pi_{\theta_{\text{old}}} \leftarrow \pi_\theta$
7:        For each $(q, a) \in \mathcal{D}_b$, sample $G$ rollouts $\{o_i\}_{i=1}^G \sim \pi_{\theta_{\text{old}}}^{\mathcal{E}}(\cdot \mid q)$
8:        For each rollout $o_i$, collect its generated-token sequence $u_i = (u_{i,1}, \ldots, u_{i,M_i})$ and contexts $\{c_{i,m}\}_{m=1}^{M_i}$
9:        Compute outcome rewards $\{r_i^{\text{O}}\}_{i=1}^G$ from final answers using Eq. 2
10:       Compute group-relative advantages $\{\widehat{A}_i\}_{i=1}^G$ by normalizing $\{r_i^{\text{O}}\}_{i=1}^G$
11:       Assign $\widehat{A}_i$ to all generated decision tokens $\{u_{i,m}\}_{m=1}^{M_i}$ in rollout $o_i$
12:       **for** GRPO update $= 1, \ldots, \mu$ **do**
13:          Update $\pi_\theta$ by maximizing the GRPO objective in Eq. 1
14:       **end for**
15:     **end for**
16: **end for**

**IGPO**

**Require:** initial policy $\pi_{\theta_{\text{init}}}$; task prompts $\mathcal{D}$; tool environment $\mathcal{E}$; rollout group size $G$; number of training steps $S$; hyperparameters $\epsilon, \beta, \gamma, \mu$

1: $\pi_\theta \leftarrow \pi_{\theta_{\text{init}}}$
2: **for** iteration $= 1, \ldots, I$ **do**
3:     $\pi_{\text{ref}} \leftarrow \pi_\theta$
4:     **for** step $= 1, \ldots, S$ **do**
5:        Sample a batch $\mathcal{D}_b$ from $\mathcal{D}$
6:        $\pi_{\theta_{\text{old}}} \leftarrow \pi_\theta$
7:        For each $(q, a) \in \mathcal{D}_b$, sample $G$ rollouts $\{o_i\}_{i=1}^G \sim \pi_{\theta_{\text{old}}}^{\mathcal{E}}(\cdot \mid q)$
8:        For each rollout $o_i = (\tau_{i,1}, \ldots, \tau_{i,T_i})$, collect $u_i = (u_{i,1}, \ldots, u_{i,M_i})$, contexts $\{c_{i,m}\}_{m=1}^{M_i}$, and interaction histories $\{C_{i,t}\}_{t=0}^{T_i-1}$
9:        Compute ground-truth confidence scores $\{s_{i,t}\}_{t=0}^{T_i-1}$ for each rollout using Eq. 3
10:       Compute information-gain rewards $\{r_{i,t}^{\text{IG}}\}_{t=1}^{T_i-1}$ using Eq. 4
11:       Compute outcome rewards $\{r_i^{\text{O}}\}_{i=1}^G$ for final answer turns using Eq. 2
12:       Normalize IG rewards and outcome rewards separately to obtain $\{\tilde{r}_{i,t}\}_{t=1}^{T_i}$ using Eq. 5
13:       Compute turn-level discounted returns $\{\tilde{R}_{i,t}\}_{t=1}^{T_i}$ using Eq. 6
14:       For each generated token $u_{i,m}$, compute its turn index $\kappa_i(m)$ and assign $\tilde{R}_{i,\kappa_i(m)}$ to $u_{i,m}$
15:       **for** IGPO update $= 1, \ldots, \mu$ **do**
16:          Update $\pi_\theta$ by maximizing the IGPO objective in Eq. 7
17:       **end for**
18:     **end for**
19: **end for**

---

# H  PROMPT TEMPLATE USED IN OUR EXPERIMENTS.

Our prompt follows the style of DeepResearcher Zheng et al. (2025b), and the same template is used for training, validation, and testing. The prompt template is shown in the Figure 9, where {today} represents the current date to ensure the relevance of the model's response. {{ tool.name }}: {{ tool.description }} indicates the available tools, while the #Rollout section controls the model's output format. The #Tools section provides the model with the tool invocation method.

```
* Today is {today}
* You are an  AI Assistant*

The question I give you is a complex question that requires a *deep research* to answer.
I will provide you with tools to help you answer the question:
{%- for tool in tools.values() %}
- {{ tool.name }}: {{ tool.description }}
{%- endfor %}

You don't have to answer the question now, but you should first think about the research plan or what to search next.
Your output format should be one of the following two formats:

# Rollout
<think>
YOUR THINKING PROCESS
</think>
<answer>
YOUR ANSWER AFTER GETTING ENOUGH INFORMATION
</answer>

  or

<think>
YOUR THINKING PROCESS
</think>
<tool_call>
YOUR TOOL CALL WITH CORRECT FORMAT
</tool_call>

You should always follow the above two formats strictly.
Only output the final answer (in words, numbers or phrase) inside the <answer></answer> tag, without any
explanations or extra information. If this is a yes-or-no question, you should only answer yes or no.

# Tools
You may call one or more functions to assist with the user query.

You are provided with function signatures within <tools></tools> XML tags:
<tools>
{\%- for tool in tools.values() \%}
   {{ '{' }}"type": "function", "function": {{ '{' }}"name": "{{ tool.name | replace("'", "") }}", "description":
"{{ tool.description }}", "parameters": {{ '{' }}"type": "object", "properties": {{tool.inputs | replace("'", "")}}, "example":
{{tool.example | replace("'", "")}}, "uniqueItems": true{{ '}}}' }}
  {\%- endfor \%}
</tools>

For each function call, return a json object with function name and arguments within <tool_call></tool_call> XML tags:
<tool_call>
{{ '{' }}"name": <function-name>, "arguments": <args-json-object>{{ '}' }}
</tool_call>
```

Figure 9: Prompt template used in our experiments.

# Task

I will provide you with a question and a reasoning trace regarding that question. Your task is to infer the answer to the question relying *solely* on the provided reasoning trace. If the answer cannot be deduced from the provided reasoning trace, you must reply with "unknown".

Question: {Question}
Reasoning Trace: {Reasoning_Traces}

#Answer

Please strictly enclose your inferred answer (or "unknown") within <answer> and </answer> tags.
Examples: <answer>Your Answer</answer> or <answer>unknown</answer>.

Figure 10: Prompt template for gemini-2.5-pro in the spurious correlations analysis.

## I CASE STUDY

Figure 11 and Figure 12 present representative examples illustrating how IGPO assigns positive rewards to useful retrieval turns and penalizes misleading ones.

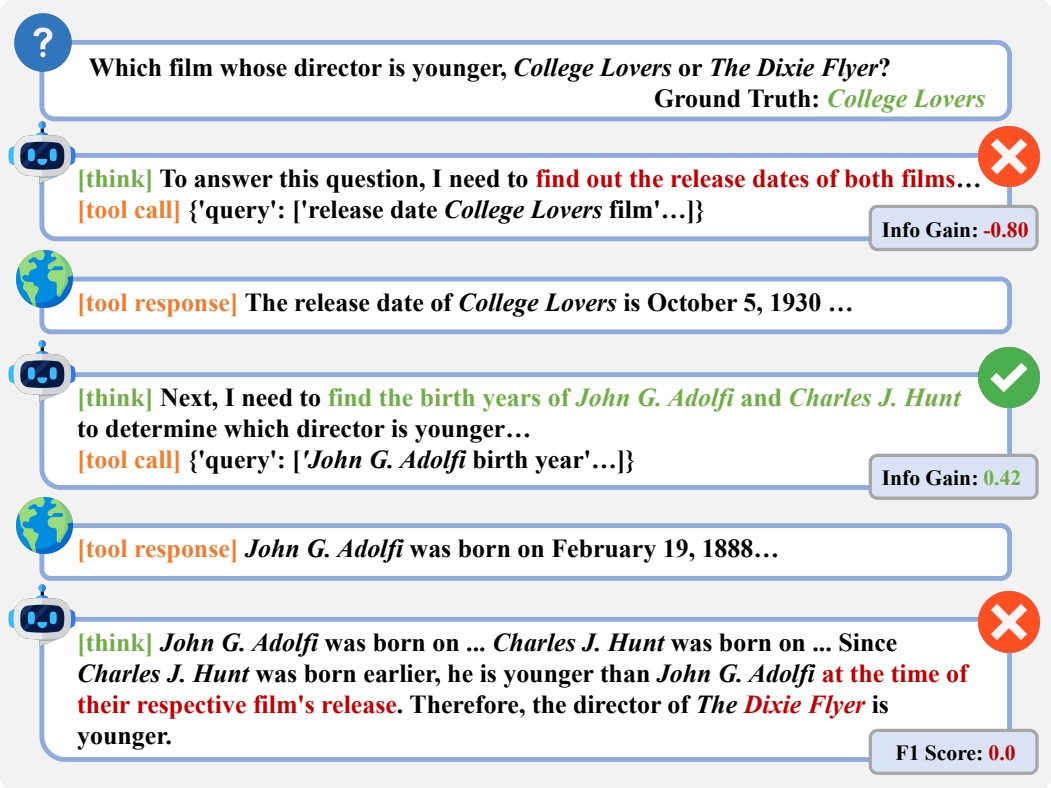

Figure 11: Case study showing a scenario where the final answer is incorrect but contains a single correct retrieval turn. IGPO provides a process reward for this turn, improving token utilization.

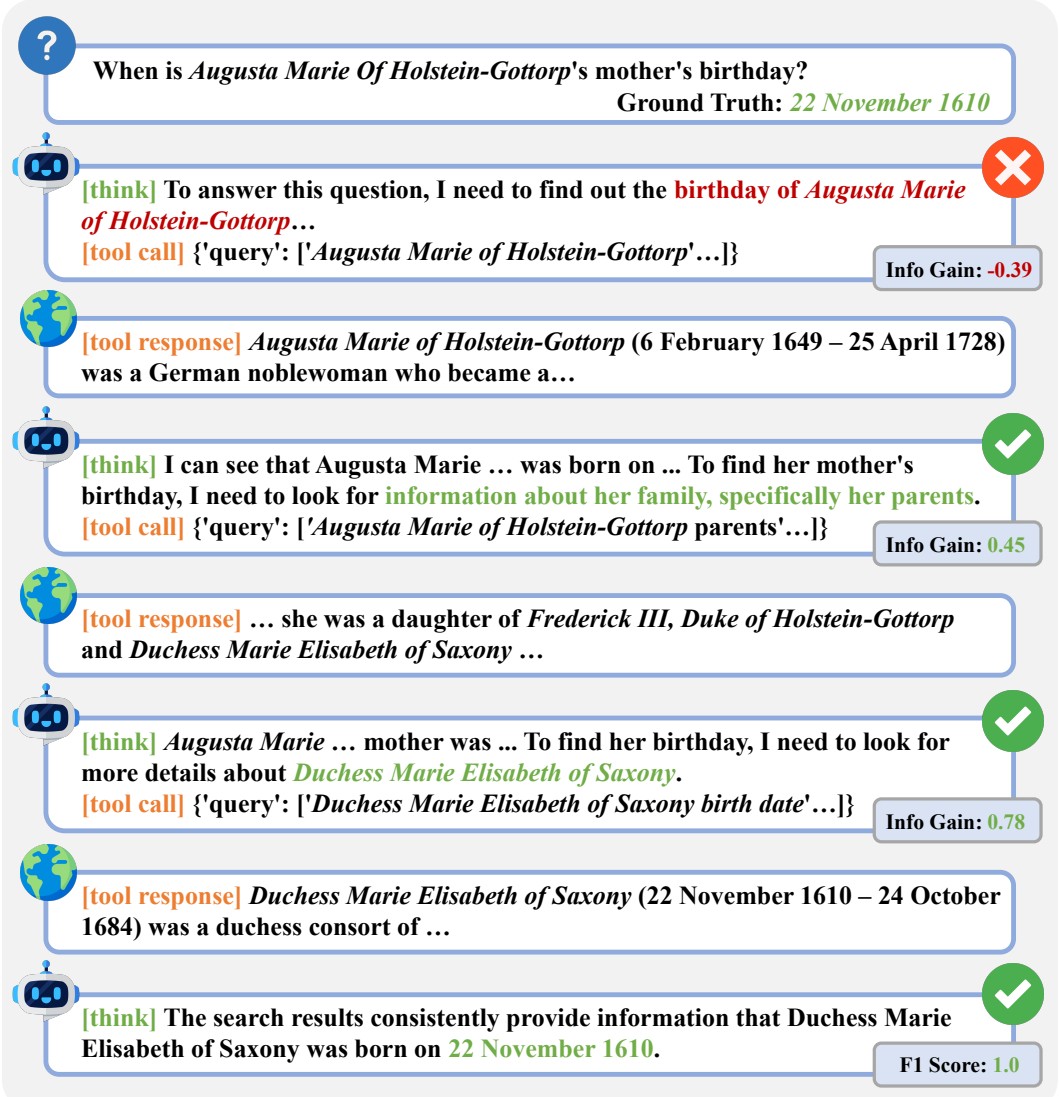

Figure 12: Case study illustrating a situation where the first round of retrieval failed, but the second and third rounds successfully located the correct evidence and produced the right answer. In this case, IGPO imposes a penalty on the erroneous retrieval in the first round.

