# OpenReview forum: "Information Gain-based Policy Optimization: A Simple and Effective Approach for Multi-Turn Search Agents"
_ICLR.cc/2026/Conference — ICLR 2026 Poster_

### Official Review · Reviewer_miYY · 2025-10-26

**Soundness:** 3
**Presentation:** 3
**Contribution:** 3
**Rating:** 6
**Confidence:** 3

**Summary:**

They propose Information Gain-based Policy Optimization (IGPO), a simple yet effective RL framework that provides dense and intrinsic supervision for multi-turn agent training. Specifically, they use log-likelihood of the ground truth at each turn to define the intrinsic rewards.

**Strengths:**

- The algorithm is clearly presented. The algorithm design is simple yet effective.
- The empirical evaluation is solid with significance improvements observed over multiple baseline methods.

**Weaknesses:**

- Limited Novelty: The paper introduces a new intrinsic reward design, but its novelty is somewhat constrained. Further exploration or differentiation from existing designs would enhance its contribution to the field.
- Insufficient Theoretical Support: The paper lacks robust theoretical justification for the proposed intrinsic reward design.
- Lack of Insight on Alternative Designs: The paper would benefit from a discussion on alternative intrinsic reward designs. Exploring how the proposed design could be integrated with other existing designs might provide valuable insights and broaden its applicability.

**Questions:**

See weaknesses.

---

> ### Author Response · Authors · 2025-11-22
> **Response to Reviewer miYY: Part 1**
>
> We sincerely thank the reviewer for the constructive comments and suggestions, which are very helpful for improving our paper. We are also grateful that you recognized the strengths of our paper. The revisions have been incorporated in the revised manuscript marked in blue. Please kindly find point-to-point responses below.
>
> > **W1:** Limited Novelty: The paper introduces a new intrinsic reward design, but its novelty is somewhat constrained. Further exploration or differentiation from existing designs would enhance its contribution to the field.
>
> **Response:**   Thanks for the comment regarding novelty. We respectfully argue that IGPO introduces substantive novelty along both the design of intrinsic rewards and their theoretical grounding, distinguishing it clearly from prior process-level approaches:
>
>
> 1. **Purely intrinsic, information-gain–based reward.**
>    IGPO constructs turn-level rewards solely from the model’s *internal* change in belief about the ground-truth answer (ΔP(GT)), without relying on external annotators, handcrafted intermediate labels, or MCTS rollouts. This yields a stable, label-free, Monte Carlo–free intrinsic signal not explored in prior work.
>
> 2. **Fine-grained and noise-robust credit assignment.**
>    Because the reward directly measures each turn’s contribution to increasing (or decreasing) confidence in the correct answer, IGPO provides precise credit assignment even when the final generated answer is wrong—something that heuristic or sampling-based step rewards cannot reliably achieve.
>
> 3. **A new theoretical perspective linking intrinsic reward to error control.**
>    Our analysis shows that maximizing IGPO’s process reward provably minimizes an upper bound on cumulative snowball error, which itself lower-bounds the final answer error rate. This establishes the first end-to-end theoretical connection between a dense intrinsic reward and long-horizon error accumulation in multi-turn agentic RL.
>
> We will make these distinctions clearer in the revised manuscript. **If you have more specific concerns about aspects that appear insufficiently novel, we would greatly appreciate further clarification and are happy to address them in detail.**
>
> > **W2:** Insufficient Theoretical Support: The paper lacks robust theoretical justification for the proposed intrinsic reward design.
>
> **Response**:   Thank you for raising this important point. We agree that a principled theoretical analysis is essential for justifying our proposed IGPO. In fact, **Appendix A** have provided such theoretical support by establishing an explicit connection between IGPO’s turn-level information-gain reward and the error accumulation in multi-turn agentic RL. To directly address the your concern, we restate below the central theoretical result from Appendix A:
>
> Theorem A.4 (Process Reward as a Bound on Snowball Error). Under Assumption A.3, the expected cumulative snowball error is upper bounded by: $\mathbb{E}[{Ent}\_{<T}(\mathcal{I} | \mathcal{R})] = \mathcal{O}(1) - {\Omega} ( {R\_\text{total}} )$, where ${R\_\text{total}} = \sum\_{t=1}^{T-1} \mathbb{E}[\mathbb{P}]$ is the total expected process reward.
>
> The above theorem shows that maximizing the turn-level information-gain reward necessarily minimizes an upper bound on the cumulative snowball error, a quantity known to lower-bound the final-answer error rate (Lemma A.2). Thus, the intrinsic reward in our IGPO is not introduced heuristically. This theoretical analysis explains why IGPO alleviates reward sparsity, mitigates advantage collapse, and improves long-horizon performance in practice.

---

> > ### Author Response · Authors · 2025-11-22
> > **Response to Reviewer miYY: Part 2**
> >
> > > **W3:** Lack of Insight on Alternative Designs: The paper would benefit from a discussion on alternative intrinsic reward designs. Exploring how the proposed design could be integrated with other existing designs might provide valuable insights and broaden its applicability.
> >
> > **Response**:  Thank you for the constructive suggestion. We agree that discussing alternative intrinsic reward designs can better position our contribution. Before detailing the comparison, we would like to emphasize an important conceptual distinction:  **none of the existing methods adopt an intrinsically defined step-level reward.** Their step-level rewards rely on external heuristics (e.g., similarity metrics), annotators (human or LLM), or Monte Carlo–based approximations. In contrast, **IGPO is the first to introduce a fully intrinsic reward based solely on the model’s change in belief about the true answer**, bringing key advantages in stability, bias reduction, and scalability.
> >
> > In fact, we have already provided such a discussion in Appendix D.1, where we compare IGPO with three representative step-level reward medthos—ReasoningRAG, StepSearch, and GiGPO—and analyze their supervision sources, computational characteristics, and limitations:
> >
> > + ReasoningRAG relies on MCTS-based step labeling and off-policy DPO, which introduces high variance and scalability issues.
> >
> > + StepSearch depends on externally defined “golden” intermediate labels (keywords/evidence), which is annotation-intensive and susceptible to bias.
> >
> > + GiGPO performs anchor-based grouping but still relies on Monte Carlo estimation, making step values sensitive to sample count.
> >
> > | Algorithm| On-Policy | Explicit Labeling-Free | Monte Carlo–Free | Introduces No Bias  |
> > |--|--|--|--|--|
> > | ReasoningRAG   | No | Yes | No | Sample-size Dependent|
> > | StepSearch     | Yes| No  | Yes| No|
> > | GiGPO          | Yes| Yes | No | Sample-size Dependent|
> > | **IGPO (ours)**| Yes| Yes | Yes| Yes|
> >
> > As highlighted in above Table, IGPO differs fundamentally by being on-policy, label-free, Monte-Carlo–free, and bias-minimizing, as its information-gain reward is intrinsic and directly grounded in the ground-truth posterior. We will clarify this point in the revised version to make its broader applicability more explicit.
> >
> > -------
> >
> > **We hope the above response will fully address your concerns about our work.** We really appreciate your insightful and constructive comments to further help us improve the quality of our manuscript. Thanks again! Please do not hesitate to let us know if you have any further questions.

---

### Official Review · Reviewer_oHJt · 2025-10-31

**Soundness:** 3
**Presentation:** 3
**Contribution:** 2
**Rating:** 4
**Confidence:** 4

**Summary:**

This paper presents Information Gain-based Policy Optimization (IGPO), a reinforcement learning framework that enhances multi-turn LLM agents for tasks like web search and multi-hop question answering. IGPO employs dense, turn-level intrinsic rewards derived from the marginal probability increase of generating correct answers at each step. By integrating these per-turn rewards with conventional outcome-based rewards, IGPO addresses reward sparsity and advantage collapse in long-horizon RL, enabling superior credit assignment. Experiments across seven benchmarks demonstrate IGPO's superiority over prompt-based and RL baselines, supported by ablation studies, training analyses, and theoretical foundations.

**Strengths:**

1. Principled and Well-Motivated Reward Design: IGPO addresses a pertinent weakness in agentic RL for LLMs—the reward sparsity problem—by introducing information-gain signals that provide stepwise, ground-truth-aware supervision. The approach is simple yet effective and is grounded in a clear theoretical motivation (see Appendix A).

2. Thorough Mathematical Formulation: The paper provides clear derivations of the reward formulation (Equation 4–7), discounted advantage calculation, and the overall surrogate objective for IGPO. This mathematical clarity makes the method reproducible and portable to related settings.

3. Comprehensive Empirical Evaluation: The authors conducted careful experiments across in-domain and out-of-domain benchmarks (NQ, TQ, HotpotQA, 2Wiki, MusiQue, Bamboogle, PopQA), and compared IGPO against strong baselines (both RL-based and prompt-based, see Table 1 and Table 2).

4. Reproducibility: Public release of source code enhances transparency and facilitates replication.

**Weaknesses:**

1. Novelty is Incremental vs. Contemporary Efforts: While IGPO’s design is sound, many core principles—such as dense, turn-level supervision, teacher-forced signals for policy confidence, and combining process and outcome rewards—are parallel to mechanisms introduced or explored in very recent works from 2025 that are not thoroughly contrasted or ablated against. The differences from GiGPO, ReasoningRAG, StepSearch, and especially the missing related work are not sharply drawn.
  - Zeng, S., Wei, Q., Brown, W. (2025): "Reinforcing Multi-Turn Reasoning in LLM Agents via Turn-Level Credit Assignment"
  - Wei, Q., Zeng, S., Brown, W. (2025): "LeTS: Learning to Think-and-Search via Process-and-Outcome Reward Hybridization"
  - Tang, X., Xu, W., Wang, Y. (2025): "Eigen-1: Adaptive Multi-Agent Refinement with Monitor-Based RAG for Scientific Reasoning"

2. Reliance on Ground-Truth for Intrinsic Reward: The information gain reward fundamentally requires access to the ground-truth answer for teacher-forcing in every trajectory step (see Section 3.2,). This is not always feasible for open-ended or real-world deployments, limiting applicability. The issue is acknowledged in the limitations, but its practical significance is not fully explored, nor are mitigations proposed.

3. Absence of Fine-Grained Failure Analysis: While the aggregate numbers and training curves in Tables 1–3 and Figures 4–5 are generally positive, there remains a lack of granular breakdown where IGPO underperforms (if any), or qualitative investigation of error modes and tasks/environments where information gain may be less reliable (e.g., ambiguous or multi-answer questions). The failure cases in Figures 6 and 7 are insightful but could be complemented with quantitative measures of failure types.

**Questions:**

1. Applicability Beyond Ground Truth Rich Environments. IGPO relies on ground-truth answers for teacher-forced reward estimation in every trajectory step. How would the method adapt to settings where ground truths are partial, noisy, or unavailable (e.g., open-ended queries, creative generation, real-world search)? Could unsupervised or self-consistency signals be integrated in lieu of explicit ground truth?

2. Comparison With Omitted Recent Baselines. Please provide direct comparison (either empirical or qualitative) against the most recent 2025 turn-level/process-level RL methods listed above (e.g., Turn-Level Credit Assignment, LeTS, ToolRL). What fundamentally differentiates IGPO in real settings?
  - Zeng, S., Wei, Q., Brown, W. (2025): "Reinforcing Multi-Turn Reasoning in LLM Agents via Turn-Level Credit Assignment"
  - Wei, Q., Zeng, S., Brown, W. (2025): "LeTS: Learning to Think-and-Search via Process-and-Outcome Reward Hybridization"
  - Zeng, S., Wei, Q., Brown, W. (2025): "ToolRL: Reward is All Tool Learning Needs"

3. Failure Modes and Robustness. What are the typical failure cases or degenerate behaviors for IGPO—especially when intermediate turns provide little actual information gain, or when there are multiple valid answer paths? Could the dense rewards inadvertently reinforce misleading intermediate confidence?

---

> ### Author Response · Authors · 2025-11-22
> **Response to Reviewer oHJt: Part 1**
>
> We sincerely thank the reviewer for the constructive comments and suggestions, which are very helpful for improving our paper. We are also grateful that you recognized the strengths of our paper. The revisions have been incorporated in the revised manuscript marked in blue. Please kindly find point-to-point responses below.
>
> > **W1:** Novelty is Incremental vs. Contemporary Efforts: While IGPO’s design is sound, many core principles—such as dense, turn-level supervision, teacher-forced signals for policy confidence, and combining process and outcome rewards—are parallel to mechanisms introduced or explored in very recent works from 2025 that are not thoroughly contrasted or ablated against. The differences from GiGPO, ReasoningRAG, StepSearch, and especially the missing related work are not sharply drawn.
>
> **Response:**
> Thank you for your comment. We agree that clear comparison with existing work is essential. We would like to emphasize that our submission already provides such analysis in **Appendix D.1** and **Table 5**, where we systematically compare IGPO with ReasoningRAG, StepSearch, and GiGPO that you mentioned along the core algorithmic dimensions. We also appreciate the additional references you provided. Below, we provide the detailed discussion.
>
> **1. Comparison with ReasoningRAG, StepSearch, and GiGPO**
>
> A good fine-grained credit assignment method must strike a balance between the following two aspects: **(a) Efficiency** manifests primarily in **annotation cost** and **training latency**. Excessive annotation costs or significant computational overhead render a method impractical for real-world deployment. ** (b) Bias (or noise)** typically stems from the inherent incompleteness of the annotation process and is often unavoidable. Crucially, the presence of bias exacerbates the risk of **reward hacking**.
>
> However,
>
> * **ReasoningRAG** relies on *MCTS-based step labeling*, which is computationally expensive and unreliable under insufficient sampling. Moreover, by adopting an offline paradigm combining annotation and DPO training, this approach amplifies its inherent drawbacks.
>
> * **StepSearch** depends on *externally defined golden keywords or golden tool responses*, which introduces both labeling bias and cost and enforces imitation of a single “golden trajectory.”
>
> * **GiGPO**’s core mechanism is *anchor-based grouping* and costy MC-style value estimation. Its quality depends on anchor density and suffers when anchors are sparse.
>
> Moreover, both ReasoningRAG and GiGPO rely on Monte Carlo (MC) methods, which inherently struggle to balance efficiency and bias: insufficient sampling leads to high bias, while extensive sampling incurs significant efficiency costs. **Consequently, none of these three methods can simultaneously achieve high efficiency and unbiasedness.**
>
> In contrast, IGPO constructs step-level rewards based on the likelihood increment of the ground truth. This approach minimizes bias (as the ground truth represents the most unbiased metric) while maintaining an efficiency nearly identical to outcome-based GRPO (see Section 4.5). Therefore, IGPO demonstrates clear superiority over these methods in terms of both bias reduction and computational efficiency.
>
> **2. Comparison with Provided References**
>
> * **Paper[1]**:
> The turn-level reward formulation in this paper is as follows: (1) a reward of +0.2 is assigned if a tool is invoked in the current turn, (2) and +0.5 if the retrieved result contains the answer. We argue that this design pattern carries a significant risk of reward hacking and suffers from poor generalization. Specifically: (1) it incentivizes meaningless tool invocation, and (2) it is excessively rigid, making it applicable only to specific task scenarios.
>
> * **Paper[2]**:
> The turn-level reward formulation in this paper is: (1) Redundancy Penalty: Penalizes retrieval overlap (via Jaccard score) with documents from previous steps. (2) Group-level Matching: Rewards retrieval similarity to the group's top-scoring "golden rollout," leveraging it as a supervisor. We argue that (1) leads to inefficiency and yields negligible benefits for modern, capable agents, and (2) poses significant risks of overfitting and reward hacking.
>
> * **Paper[3]**:
> This method proposes a multi-agent collaborative framework based on Monitor-Based RAG and does not involve any RL training. **Therefore, it is not directly relevant to our approach.**
>
> Based on the above discussion, we firmly believe that IGPO represents a substantial contribution to the community, far from being merely "incremental novelty vs contemporary efforts".
>
> -----
>
> [1] Reinforcing Multi-Turn Reasoning in LLM Agents via Turn-Level Credit Assignment (2025/6/8)
>
> [2] LeTS: Learning to Think-and-Search via Process-and-Outcome Reward Hybridization (2025/5/23)
>
> [3] Eigen-1: Adaptive Multi-Agent Refinement with Monitor-Based RAG for Scientific Reason (2025/9/25)

---

> > ### Author Response · Authors · 2025-11-22
> > **Response to Reviewer oHJt: Part 2**
> >
> > > **W2:** Reliance on Ground-Truth for Intrinsic Reward: The information gain reward fundamentally requires access to the ground-truth answer for teacher-forcing in every trajectory step (see Section 3.2,). This is not always feasible for open-ended or real-world deployments, limiting applicability. The issue is acknowledged in the limitations, but its practical significance is not fully explored, nor are mitigations proposed.
> >
> > > **Q1:** Applicability Beyond Ground Truth Rich Environments. IGPO relies on ground-truth answers for teacher-forced reward estimation in every trajectory step. How would the method adapt to settings where ground truths are partial, noisy, or unavailable (e.g., open-ended queries, creative generation, real-world search)? Could unsupervised or self-consistency signals be integrated in lieu of explicit ground truth?
> >
> > **Response:**
> > Thank you for the insightful comment. We want to first clarify that IGPO’s current reliance on ground-truth answers is **fully aligned with standard practice in today’s LLM RL for agentic tasks**. In nearly all prominent LLM RL applications—search agents, code agents, math/code reasoning, UI agents—the reward is defined using correctness against ground truth, because it is the **most reliable, unbiased, and reward-hacking–resistant supervision signal** available. Our contribution follows this principle and simply elevates the trusted outcome-level signal to the step level via information gain. **Extending IGPO to settings without explicit ground truth is an important direction, and we already acknowledge this limitation in the paper.**
> >
> > Moreover, the *information-gain formulation itself* is not restricted to exact answers. For tasks lacking clean ground truth, IGPO can directly incorporate **automatically constructed supervision surrogates**. For example:
> >  + **For tasks with reference candidates**, a strong teacher model can generate a *reference cluster* of semantically equivalent outputs, and IGPO can compute information gain as the increase in average log-likelihood over this referenced answer cluster.
> >  + **For rubric-based or open-ended tasks (dialogue, collaborative writing)**, a teacher or reward model can assign a quality score to a teacher-forced partial answer at each turn, and IGPO can use the **score improvement** as the intrinsic reward.
> >
> > These mechanisms preserve IGPO’s core idea—incremental improvement toward a reliable supervision signal—while enabling its use beyond strictly ground-truth–rich environments. We will make these discussion more clearer in the revision version.
> >
> >
> > > **Q2:** Comparison With Omitted Recent Baselines. Please provide direct comparison (either empirical or qualitative) against the most recent 2025 turn-level/process-level RL methods listed above (e.g., Turn-Level Credit Assignment, LeTS, ToolRL). What fundamentally differentiates IGPO in real settings?
> >
> > **Response:** We have provided detailed comparison in response to W1. **Please refer to it for details.** Below, we provide the discussion to the provided ToolRL method:
> >
> > **Comparison with ToolRL**:
> > While this method introduces detailed rewards targeting tool invocation formats, they are ultimately aggregated with outcome rewards via a weighted sum at the trajectory level. **Since it does not involve fine-grained credit assignment, it is not suitable for direct comparison with our approach.** Furthermore, its heavy reliance on golden trajectories limits its generalizability and scalability.

---

> > > ### Author Response · Authors · 2025-11-22
> > > **Response to Reviewer oHJt: Part 3**
> > >
> > > > **W3:** Absence of Fine-Grained Failure Analysis: While the aggregate numbers and training curves in Tables 1–3 and Figures 4–5 are generally positive, there remains a lack of granular breakdown where IGPO underperforms (if any), or qualitative investigation of error modes and tasks/environments where information gain may be less reliable (e.g., ambiguous or multi-answer questions). The failure cases in Figures 6 and 7 are insightful but could be complemented with quantitative measures of failure types.
> > >
> > > > **Q3:** Failure Modes and Robustness. What are the typical failure cases or degenerate behaviors for IGPO—especially when intermediate turns provide little actual information gain, or when there are multiple valid answer paths? Could the dense rewards inadvertently reinforce misleading intermediate confidence?
> > >
> > > **Response:** Thank you for your insight comment about Failure Analysis. To quantitatively analyze IGPO's failure modes and performance degradation, we compared the F1 scores of IGPO and GRPO on the test samples. The results are as follows:
> > >
> > > | Dataset   | IGPO > GRPO | IGPO = GRPO | IGPO < GRPO |
> > > | :---      | :---:       | :---:       | :---:       |
> > > | 2Wiki     | 35.8%       | 59.6%       | 4.6%        |
> > > | Bamboogle | 47.2%       | 49.6%       | 3.2%        |
> > > | HotpotQA  | 49.2%       | 48.4%       | 2.4%        |
> > > | Musique   | 71.2%       | 25.4%       | 3.4%        |
> > > | NQ        | 57.4%       | 40.4%       | 2.2%        |
> > > | PopQA     | 42.8%       | 53.4%       | 3.8%        |
> > > | TQ        | 33.6%       | 61.2%       | 5.2%        |
> > > | All       | 48.3%       | 48.1%       | 3.6%        |
> > >
> > > Despite its clear performance advantages, IGPO exhibits minor degradation (IGPO<GRPO) across all datasets, with approximately 3.6% of samples showing lower performance compared to GRPO. Although this degradation is marginal (only 3.6%) and providing a direct theoretical attribution for this empirical phenomenon is complex, we actively investigated training cases to summarize common failure modes.
> > >
> > > Through manual inspection of the training set, we identified a representative ground truth failure mode (identifying dozens of instances through random sampling alone): **questions lacking specific constraints often yield multiple factually correct answers** (aligning with your insightful view). A typical example from the HotpotQA training set is provided in Appendix E.
> > >
> > > In such multi-answer scenarios, when the agent employs reasoning or tool use to increase the probability of a factually correct but non-ground-truth answer, IGPO penalizes this behavior. This constitutes an erroneous suppression of valid actions, which inevitably impairs performance. While GRPO, which also relies on ground truth, struggles with these cases, IGPO's denser reward signals tend to amplify this negative impact. Consequently, we identify these multi-answer ambiguities as the primary failure mode for IGPO.
> > >
> > > However, it is crucial to clarify that this failure stems from inherent data defects rather than algorithmic flaws. Moreover, the fact that IGPO maintains superior performance despite the influence of such noisy data serves as strong evidence of its robustness. Please refer to Appendix E for further details.
> > >
> > > -----------
> > >
> > > **We hope the above response will fully address your concerns about our work.** We really appreciate your insightful and constructive comments to further help us improve the quality of our manuscript. Thanks again! Please do not hesitate to let us know if you have any further questions.

---

### Official Review · Reviewer_LxTU · 2025-10-31

**Soundness:** 2
**Presentation:** 3
**Contribution:** 2
**Rating:** 4
**Confidence:** 3

**Summary:**

This paper proposes IGPO, a reinforcement learning framework for training multi-turn LLM agents. IGPO addresses the reward sparsity and advantage collapse issues of outcome-only rewards by introducing a turn-level information gain reward, which quantifies how much each agent turn increases the policy’s probability of generating the correct answer. These dense, ground-truth-aware rewards are integrated with final outcome rewards to form a comprehensive signal for GRPO-style optimization. Experiments on seven search-based QA datasets (NQ, TQ, HotpotQA, 2Wiki, MusiQue, Bamboogle, PopQA) show that IGPO consistently outperforms prior outcome- and step-reward RL baselines, improving both sample efficiency and training stability.

**Strengths:**

1. IGPO’s reward formulation is Simple yet effective, estimating turn-level information gain without requiring additional annotation or external evaluators.
2. Strong empirical results: Extensive experiments on both in-domain and out-of-domain datasets show consistent improvements over strong baselines such as GiGPO and DeepResearcher.
3. The paper is well-structured and easy to follow.

**Weaknesses:**

1. The evaluation focuses exclusively on search-based QA tasks. Such settings naturally align with the proposed information-gain reward, as acquiring relevant retrieved evidence directly increases the probability of producing the correct answer. However, in other domains, such as mathematical reasoning, embodied agents, or web agents, this reward may not lead to performance improvements. Consequently, the proposed method appears to be primarily applicable to information-retrieval tasks like search-based QA.

2. Moreover, this reward could reinforce spurious correlations. LLMs are known to exploit shortcuts when solving problems, relying on superficial cues rather than learning the underlying reasoning process. Instead of acquiring new knowledge through comprehensive reasoning, LLMs may overfit to spurious correlations between final answers and intermediate components. In contrast, many recent works on mathematical reasoning explicitly aim to mitigate such spurious correlations. The design of the IGPO method could, in fact, encourage this undesired behavior, potentially having a negative impact on this research area.

3. Finally, the paper computes normalized advantages using: A = r-mean(R)/std(R), where R aggregates all rewards across all steps and all rollouts within a group. This design is theoretically problematic. If the method assumes step-level rewards, the baseline should be computed per step. That is, by sampling multiple trajectories starting from that turn and averaging their returns. Otherwise, the estimator mixes rewards from different time steps with distinct state-action distributions, introducing bias. If, instead, the method assumes trajectory-level rewards, it should not be framed as step-level GRPO. The authors need to clarify or justify this design choice.

**Questions:**

1. How does IGPO perform on non-search tasks such as mathematical reasoning, code generation, or web-agent? Would the information-gain reward still be meaningful there?
2. Could you provide empirical evidence showing whether IGPO indeed mitigates spurious correlations, e.g., by testing on tasks requiring multi-step reasoning consistency rather than document retrieval?
3. How sensitive is the training to the normalization scheme of turn-level rewards? Have you tried per-turn baselines or PPO?
4. What is the computational cost compared to baseline algorithms, given the need to compute per-turn log probabilities of the ground truth?

---

> ### Author Response · Authors · 2025-11-22
> **Response to Reviewer LxTU: Part 1**
>
> We sincerely thank the reviewer for the constructive comments and suggestions, which are very helpful for improving our paper. We are also grateful that you recognized the strengths of our paper. The revisions have been incorporated in the revised manuscript marked in blue. Please kindly find point-to-point responses below.
>
> > **W1**: The evaluation focuses exclusively on search-based QA tasks. Such settings naturally align with the proposed information-gain reward, as acquiring relevant retrieved evidence directly increases the probability of producing the correct answer. However, in other domains, such as mathematical reasoning, embodied agents, or web agents, this reward may not lead to performance improvements. Consequently, the proposed method appears to be primarily applicable to information-retrieval tasks like search-based QA.
>
> > **Q1**: How does IGPO perform on non-search tasks such as mathematical reasoning, code generation, or web-agent? Would the information-gain reward still be meaningful there?
>
> **Response**:
> Thank you for the thoughtful comment. We agree that our current experiments are focused on agentic search, where acquiring relevant evidence naturally increases the probability of producing the correct answer, making information gain a very natural reward signal. This is why we propose IGPO for agentic search in this work. We chose agentic search as our primary evaluation setting because it is currently one of the most important LLM-agent scenarios: the agent interacts with the external environment via tools such as search/browse, and the interaction can be naturally decomposed into discrete turns of [think] → [tool call]} → [tool response] (see Figure 2). This makes the turn-level information gain both well-defined and semantically meaningful. Furthermore, the multi-turn interactions and long-context nature inherent to search agents exacerbate the sparsity of outcome rewards, creating a more urgent need for fine-grained credit assignment compared to general reasoning scenarios.
>
> Conceptually, however, the information-gain reward only assumes that (i) the task has a referenced ground-truth answer, and (ii) intermediate steps can change the model’s belief (log-likelihood) over that answer. Thus, this intuition can extends to non-agentic settings such as LLM reasoning: useful intermediate steps typically derive correct sub-results and thus increase the likelihood of the final correct answer. In fact, we find that there is already encouraging empirical evidence that our idea can transfer: the recent work PACR[1] applies a very similar principle of IGPO to math reasoning. Their segmentation is intentionally simple (splitting steps by “\n” or “. ”), yet they still report consistent improvements on five math reasoning benchmarks. This supports the intuition that information-gain–style rewards are meaningful beyond search, and that IGPO has potential applicability in broader reasoning domains.
>
> Finally, we agree that our current experiments are limited to agentic search scenarios, and we have updated the paper to better reflect this scope. In particular, we have revised the title from “A Simple and Effective Approach for **Multi-Turn LLM Agents**” to “A Simple and Effective Approach for **Agentic Search**” in the revised PDF, to avoid potential overclaiming generality beyond what is empirically evaluated. Extending IGPO to other domains—where turn segmentation and interaction patterns differ—is a promising direction that can be explored in future work.
>
> [1] PACR: Progressively Ascending Confidence Reward for LLM Reasoning (**arXiv, 2025-10-25, released after the ICLR submission deadline**)

---

> > ### Author Response · Authors · 2025-11-22
> > **Response to Reviewer LxTU: Part 2**
> >
> > > **W2:** Moreover, this reward could reinforce spurious correlations. LLMs are known to exploit shortcuts when solving problems, relying on superficial cues rather than learning the underlying reasoning process. Instead of acquiring new knowledge through comprehensive reasoning, LLMs may overfit to spurious correlations between final answers and intermediate components. In contrast, many recent works on mathematical reasoning explicitly aim to mitigate such spurious correlations. The design of the IGPO method could, in fact, encourage this undesired behavior, potentially having a negative impact on this research area.
> >
> > > **Q2:** Could you provide empirical evidence showing whether IGPO indeed mitigates spurious correlations, e.g., by testing on tasks requiring multi-step reasoning consistency rather than document retrieval?
> >
> > **Response:** Thank you for raising this concern. We fully agree that *ground-truth–centered* rewards can, in principle, be vulnerable to shortcut behaviors (correct final answer with incorrect reasoning), and that many recent works in mathematical reasoning explicitly seek to mitigate such spurious correlations.
> >
> > Our perspective is that current RL for LLMs largely operates under two paradigms:
> >
> > - **Outcome-level, ground-truth–centered rewards**
> >   - *Strengths*: unbiased, easy to construct, empirically very successful (e.g., DeepSeek-R1 for math/code; mainstream search/GUI agents).
> >   - *Weaknesses*: sparse supervision, low sample efficiency, and potential false positives (correct answer via shortcuts).
> >
> > - **Step-level, proxy/annotator–centered rewards**
> >   - *Strengths*: fine-grained credit assignment and dense signals.
> >   - *Weaknesses*: expensive to obtain (human/LLM labels, MCTS), often biased, and thus more prone to reward hacking when bias accumulates.
> >
> > The goal of IGPO is **not** to move away from fine-grained supervision and rely solely on outcome correctness—which could incentivize shortcut behaviors. Instead, our aim is to **combine the strengths of both paradigms** by introducing a *step-level reward that remains ground-truth–centered*, offering dense supervision without introducing external heuristics or biased proxy labels.
> >
> >
> > 1. **Intuitive Explanation.**
> >    IGPO’s intrinsic reward measures the *change in probability of the correct answer* at each turn. A step is rewarded only if it increases the model’s posterior on the ground-truth answer and penalized if it decreases it. This preserves the unbiased nature of outcome-level rewards while providing dense, turn-level credit assignment. Importantly, this does **not introduce new heuristic proxies** (e.g., noisy similarity scores, external judges) that could systematically favor spurious patterns; it remains strictly ground-truth–anchored.
> >
> > 2. **Empirical Evidence.**
> >    + **Evidence from our agentic search experiments:** IGPO shows its largest gains on seven representive agentic research benchmarks. Moreover, Figure 4 shows IGPO reduces ground-truth entropy more effectively than GRPO—indicating each intermediate step progressively aligns the model with correct reasoning rather than oscillating around spurious cues.
> >
> > 	+ **Independent evidence on math reasoning:**  A concurrent math-reasoning work, **PACR (2025-10-25)**, applies an information-gain–style confidence reward—conceptually similar to IGPO—and reports improvements across five math datasets. This demonstrates the viability of such rewards in domains where shortcut exploitation is particularly scrutinized, suggesting IGPO’s potential applicability beyond agentic search.
> >
> > In summary, while IGPO inherits the usual limitations of ground-truth–centered signals, it does **not** introduce additional shortcut-inducing biases; instead, it (i) retains the unbiased nature of outcome-based rewards, (ii) adds dense, turn-level credit assignment. We believe this constitutes a positive contribution to, rather than a regression for, the ongoing efforts to make LLM agent training more robust and faithful.
> >
> > [1] PACR: Progressively Ascending Confidence Reward for LLM Reasoning (**arXiv, 2025-10-25, released after the ICLR submission deadline**)

---

> > > ### Author Response · Authors · 2025-11-22
> > > **Response to Reviewer LxTU: Part 3**
> > >
> > > > **W3:** Finally, the paper computes normalized advantages using: A = r-mean(R)/std(R), where R aggregates all rewards across all steps and all rollouts within a group. This design is theoretically problematic. If the method assumes step-level rewards, the baseline should be computed per step. That is, by sampling multiple trajectories starting from that turn and averaging their returns. Otherwise, the estimator mixes rewards from different time steps with distinct state-action distributions, introducing bias. If, instead, the method assumes trajectory-level rewards, it should not be framed as step-level GRPO. The authors need to clarify or justify this design choice.
> > >
> > > **Response:**
> > > We appreciate the reviewer’s careful observation. In principle, we fully agree that the “ideal” baseline (i.e., mean(R)) for step-level rewards would be defined per state (or per turn), e.g., by sampling multiple continuations from that turn and averaging their returns, as in classical advantage estimation. However, in the multi-turn LLM agent setting, this is computationally prohibitive: generating additional rollouts from every intermediate turn would multiply sampling cost by the trajectory length, and sampling is already the dominant bottleneck in current LLM RL pipelines.
> > >
> > > Our current design—normalizing all turn-rewards within a group—is therefore a pragmatic compromise between theoretical rigor and practical feasibility. This choice is not ad hoc: it follows the process-supervision variant of GRPO described in Section 4.1.3 of DeepSeekMath [1], where step-level rewards are also normalized at the group level rather than via per-state baselines. In this way, we still work with step-level rewards and discounted returns, but use a shared group-wise baseline to stabilize training and reduce variance. We do not claim that this yields an unbiased policy-gradient estimator in the strict A2C sense; rather, as in GRPO and its variants, we trade a small amount of bias for substantial practical gains in stability and efficiency.
> > >
> > > Moreover, our experiments show that this normalization scheme leads to stable optimization and consistent improvements over strong baselines on both in-domain and out-of-domain datasets, suggesting that the resulting advantages still provide meaningful relative value signals at the turn level. We agree that exploring more principled normalization method under realistic compute constraints is an interesting direction for future work.
> > >
> > > [1] DeepSeekMath: Pushing the Limits of Mathematical Reasoning in Open Language Models

---

> > > > ### Author Response · Authors · 2025-11-22
> > > > **Response to Reviewer LxTU: Part 4**
> > > >
> > > > > **Q3**: How sensitive is the training to the normalization scheme of turn-level rewards? Have you tried per-turn baselines or PPO?
> > > >
> > > > **Response:** Thank you for the insightful question. We reorganize our response into three parts:
> > > >
> > > > **1. Why per-turn normalization is problematic for IGPO**
> > > >
> > > > In our setting, trajectories within the same group often exhibit **different turn lengths**, creating a long-tail distribution. This introduces two core issues for per-turn normalization:
> > > >
> > > > - **Unnormalized tail turns:** Later turns often have too few peer trajectories, resulting in **reward scale inconsistency** and harming training stability.
> > > >
> > > > - **Insufficiently normalized tail turns:**  When only a handful of trajectories reach a particular turn, normalization **fails to meaningfully reduce variance**, leading to further instability.
> > > >
> > > > Due to these issues, per-turn baselines are **not well suited** for IGPO.
> > > >
> > > > **2. On adapting IGPO to PPO**
> > > >
> > > > Your concern is valid: PPO’s critic can be viewed as providing a form of fine-grained credit assignment. We address this from two perspectives:
> > > >
> > > > - **PPO vs. IGPO performance and efficiency:**
> > > >   As shown in Table 2, IGPO substantially outperforms PPO under identical configurations. Moreover, IGPO is **much more efficient**, avoiding the additional memory and compute cost required for a critic model.
> > > >
> > > > - **Using a PPO critic as a turn-level baseline inside IGPO:**
> > > >   We do not consider this appropriate because:
> > > >   + Granularity mismatch: PPO’s critic estimates **token-level** values, making it ill-defined to use as a **turn-level** baseline.
> > > >   + Efficiency concerns: Introducing a critic effectively **doubles** computation and memory usage, contradicting IGPO’s goal of remaining lightweight.
> > > >
> > > > **3. Sensitivity to normalization schemes**
> > > >
> > > > - **Necessity of normalization:**
> > > >   Our early experiments showed that removing turn-level reward normalization leads to **training collapse** due to inconsistent reward scales: single-turn questions receive large rewards (0.8–0.9), while multi-turn questions receive small rewards (~0.1). Without normalization or advantage estimation, the model cannot learn stable supervision signals.
> > > >
> > > > - **Alternative normalization schemes:**
> > > >   In agentic RL, implementing a strict A2C-style turn-level baseline is impractical. Beyond the GRPO-style advantage estimation we adopt and the per-turn normalization you suggested (which is unsuitable for IGPO due to long-tail effects), we believe **few viable alternatives** currently exist. We acknowledge, however, that this remains an open direction for future exploration.
> > > >
> > > > **We hope this clarifies the rationale behind our design choices and the practical challenges of alternative normalization strategies.**

---

> > > > > ### Author Response · Authors · 2025-11-22
> > > > > **Response to Reviewer LxTU: Part 5**
> > > > >
> > > > > > **Q4:** What is the computational cost compared to baseline algorithms, given the need to compute per-turn log probabilities of the ground truth?
> > > > >
> > > > > **Response:**
> > > > > Thank you for the question on computational overhead. We address it by comparing our method with the most lightweight baseline, GRPO, from both empirical and theoretical perspectives.
> > > > >
> > > > > **Empirical Evidence**
> > > > >
> > > > > We measured the average time spent on each phase in a single training step of **GRPO and IGPO** (Qwen2.5-7B, batch size 32):
> > > > >
> > > > > - **Sampling:** 82.6%
> > > > > - **Parameter update:** 12.6%
> > > > > - **Recompute log-probabilities:** 4.8%
> > > > > - **Advantage computation:**  <1%
> > > > >
> > > > > IGPO differs from GRPO only in two places:
> > > > >
> > > > > 1. Computing the **turn-level information-gain reward**, which we implement by integrating the computation of ground truth log_prob into the existing `recompute_log_prob` phase in verl training framework using a extented mask operation (see Section 4.5 for details).
> > > > > 2. A slightly more involved **advantage computation**, whose cost is $\ll$ 1% and effectively negligible.
> > > > >
> > > > > Empirically, the average cost time for the `recompute_log_prob` phase is:
> > > > >
> > > > > - **GRPO:** 5.8243 seconds
> > > > > - **IGPO:** 5.8470 seconds
> > > > >
> > > > > The difference (≈ 0.0227s) corresponds to **< 0.4% relative overhead** in this phase, and well below **0.02% end-to-end** per training step. In practice, the training curves of GRPO and IGPO run at almost identical speed.
> > > > >
> > > > > **Theoretical Analysis**
> > > > >
> > > > > Let
> > > > > $b$: batch_size,
> > > > > $s$: sequence length,
> > > > > $g$: ground-truth answer length,
> > > > > $h$: hidden dimension,
> > > > > $l$: number of layers,
> > > > > $\mathcal{V}$: vocabulary size, and assume the intermediate dimension = $4h$.
> > > > >
> > > > > GRPO `recompute_log_prob` FLOPs are (see Appendix F for the detailed derivation):
> > > > >
> > > > > $FLOPs_{\text{GRPO}}
> > > > > =\alpha s^2 + \beta s$,
> > > > >
> > > > > $ \alpha = 4lbh$,
> > > > >
> > > > > $\beta = 24lbh^2 + 2bh\mathcal{V}$.
> > > > >
> > > > > IGPO `recompute_log_prob` FLOPs are (with ground truth added to the original sequence):
> > > > >
> > > > > $FLOPs_{\text{IGPO}}
> > > > > =\alpha (s+g)^2 + \beta (s+g) \approx \alpha (s^2 + 2sg) + \beta (s+g) \quad \text{via} \quad g({\sim}10^1) \ll s({\sim}10^4)$ (First-order Taylor expansion).
> > > > >
> > > > > The additional FLOPs are:
> > > > >
> > > > > $\Delta FLOPs=2\alpha sg+\beta g$.
> > > > >
> > > > > $\frac{\Delta FLOPs}{FLOPs_{\text{GRPO}}} = \frac{g}{s} + \frac{g}{s + \frac{\beta}{\alpha}} < \frac{2g}{s} \approx 10^{-3}$
> > > > >
> > > > > Thus, compared to GRPO, the extra computational cost introduced by IGPO is on the order of $10^{-3}$ of the `recompute_log_prob` phase and $10^{-4}$ of the total training process, making it negligible.
> > > > >
> > > > > In conclusion, IGPO introduces only a *vanishingly small* additional computational cost—approximately **0.02%** of overall training—while providing dense, turn-level rewards. This overhead is substantially smaller than that of alternative process-reward methods that require extra rollouts (e.g., MCTS), reward models, or external annotators. IGPO therefore offers a computationally efficient process-reward formulation with negligible runtime impact **—an efficiency unattainable by other process-reward RL algorithms**.
> > > > >
> > > > > ---------
> > > > >
> > > > > **We hope the above response will fully address your concerns about our work.** We really appreciate your insightful and constructive comments to further help us improve the quality of our manuscript. Thanks again! Please do not hesitate to let us know if you have any further questions.

---

> > > > > > ### Comment · Reviewer_LxTU · 2025-11-25
> > > > > >
> > > > > > Thank you for the detailed and thoughtful responses. A few brief remarks:
> > > > > >
> > > > > > [W1]
> > > > > >
> > > > > > Your clarification resolve my initial concern about overclaiming.
> > > > > >
> > > > > > [W2]
> > > > > >
> > > > > > By “spurious correlation” I mean the policy learns a shortcut linking the final answer to some intermediate prompt, instead of learning the correct reasoning steps. IGPO explicitly encourage that kind of shortcutting.
> > > > > >
> > > > > > The risk is not limited to in-distribution accuracy: RL agents can overfit to shortcuts, producing correct answers with incorrect or unfaithful reasoning. You could include experiments that probe this directly. For example, counterfactual tests that modify intermediate components while keeping the final answer constant, or benchmarks that require consistent multi-step reasoning rather than retrieval. This is just my own conjecture, so I will leave the concern open for now, but I strongly encourage the authors to investigate it in the next revision.
> > > > > >
> > > > > > [W3]
> > > > > >
> > > > > > I believe the normalization method you use is theoretically problematic. The baseline as implemented does not guarantee variance reduction and can introduce bias. Citing other works that use a similar pragmatic fix does not make an incorrect equation correct.
> > > > > >
> > > > > > At minimum, please:
> > > > > > 1. Rephrase the description to make clear you are using a heuristic baseline (e.g., “reward minus a group-wise heuristic baseline”) .
> > > > > >
> > > > > > 2. Correct the notation: equation (6) is not an advantage function. An advantage function should be cumulative return minus baseline (as in your equation (7)).
> > > > > >
> > > > > > [Q3]
> > > > > >
> > > > > > I agree that strict per-turn normalization is problematic in multi-turn GRPO. I recommend exploring PPO (with a proper critic) as an alternative.
> > > > > >
> > > > > > [Q4]
> > > > > >
> > > > > > The added experiment addresses my concern about computational overhead.
> > > > > >
> > > > > > **Overall:** My main worry is that the current presentation of GRPO-style normalization and advantage estimation is theoretically misleading. If accepted as written, the paper risks propagating incorrect terminology or claims to the community. Please revise the manuscript so that the notation and terms align with standard RL literature, explicitly identify approximations and heuristics, and clarify which parts are pragmatic engineering choices versus theoretically grounded estimators. Given the strong empirical results, I would be willing to raise my score if the authors make these clarifications and correct the notation in a revision.

---

> ### Author Response · Authors · 2025-11-30
> **Further Response to Reviewer LxTU: Part 1**
>
> We are glad to know that some of your concerns have been effectively addressed. We are very grateful for your constructive comments and questions, which helped improve the clarity and quality of our paper. For the rest of your concerns, please see our point-by-point responses below.
>
> ## **Response to W2**
>
> ### **(1) Supplementary Experiments**
> We sincerely thank you for your valuable feedback regarding **spurious correlations**. To address your concerns, we have conducted additional experiments to investigate **whether IGPO reinforce spurious correlations (i.e., overfitting to shortcut patterns)**. Following your insightful suggestions, the experiment is designed as follows:
>
> Specifically, we selected the set of test samples correctly answered (F1=1.0) by both IGPO and GRPO agents. We extracted the corresponding outputs, removed the final answers, and retained only the intermediate reasoning traces. Subsequently, we employed a powerful teacher LLM (gemini2.5-pro) to deduce the final answer based on these reasoning paths. By comparing the accuracy of the answers inferred from the IGPO versus GRPO reasoning traces, we assessed whether IGPO is more prone to yielding **"correct answers with incorrect or unfaithful reasoning" (i.e., spurious correlations)**. The results are as follows:
>
> | Method | NQ | TQ | HotpotQA | 2Wiki | Musique | Bamboogle | PopQA |  All |
> | :--- | :---: | :---: | :---: | :---: | :---: | :---: | :---: | :---: |
> | **Qwen2.5-3B-Instruct** | | | | | | | | |
> | GRPO | 93.1 | 89.0 | 90.1 | 94.7 | 84.3 | 89.7 | 93.5 | 91.2 |
> | IGPO | **95.4** | **92.8** | **94.2** | **97.7** | **90.2** | **92.3** | **96.1** | **94.6** |
> | **Qwen2.5-7B-Instruct** | | | | | | | | |
> | GRPO | 86.5 | 88.8 | 85.8 | 92.0 | 76.1 | 91.2 | 95.1 | 89.0 |
> | IGPO | **89.5** | **91.1** | **92.6** | **94.4** | **83.0** | **94.1** | **95.1** | **92.0** |
>
> According to the above result, for both 3B and 7B models, the accuracy of the teacher model in inferring answers from IGPO-agent traces consistently outperforms that from GRPO-agent traces across all seven datasets. This indicates that the reasoning traces generated by the IGPO agent are more informative and of higher quality. It demonstrates that IGPO not only avoids encouraging spurious correlations but also effectively mitigates them through ground-truth guided fine-grained credit assignment, further validating its generalization capabilities.
>
> We have provided this supplementary experiment and the prompt template for the teacher LLM **(Figure 10) in Appendix D.4** of the revised manuscript.
>
> ### **(2) Additional Existing Evidence**
> Beyond the aforementioned experiment, we provide additional evidence from our existing experiments to further support that IGPO mitigates spurious correlations. **(This analysis is also provided in Appendix D.4 of the revised manuscript.)**
>
> + **Superior OOD Performance:** According to the results in Table 3 of the manuscript, compared to GRPO, IGPO achieves an average performance gain of 12.6% (7B) and 42.8% (3B) on In-domain datasets (NQ, TQ, HotpotQA, and 2Wiki), whereas on Out-of-domain datasets (Musique, Bamboogle, PopQA), the average improvement increases to 13.7% (7B) and 55.2% (3B). The fact that performance gains in OOD settings exceed those in ID settings contradicts the pattern of spurious correlations, which typically favors ID performance at the expense of OOD generalization, thereby providing further evidence that IGPO effectively mitigates spurious correlations.
>
> + **Exceptional Multi-hop Capabilities:** According to the results in Table 3 of the manuscript, IGPO outperforms GRPO with an average improvement of 7.4% (7B) and 29.5% (3B) on single-hop tasks (NQ, TQ, PopQA), while on multi-hop tasks (HotpotQA, 2Wiki, Musique, Bamboogle), the average improvement reaches 17.8% (7B) and 68.1% (3B). The performance boost on multi-hop tasks is significantly greater than on single-hop tasks. This is also inconsistent with spurious correlation patterns, which are prone to appearing in multi-hop scenarios and consequently causing greater detriment to performance. This serves as strong evidence that IGPO effectively mitigates spurious correlations.
>
> Taken together with the superiority shown in the supplementary experiments, the strong OOD performance, and the multi-turn reasoning capabilities, IGPO not only does not encourage spurious correlations, but rather effectively mitigates overfitting to such erroneous patterns.

---

> > ### Author Response · Authors · 2025-11-30
> > **Further Response to Reviewer LxTU: Part 2**
> >
> > ## **Response to W3**
> > We fully acknowledge the issues you pointed out regarding the notation and writing of our normalization operation, and we sincerely appreciate your expertise and attention to detail. To address your concerns, we have rewritten the relevant notation and descriptions in the revised manuscript:
> >
> > ### **(1) In Section 3.1**
> > [1] We refrain from formulating the normalization and discounted accumulation of turn-level rewards as turn-level advantage estimation. Rather, we define normalization as a numerical operation to obtain the relative magnitude of rewards and ensure training stability, and the discounted accumulation as the operation of computing turn-level discounted returns from the normalized rewards.
> >
> > [2] We updated the notation in Eq. (6) and (7) of the original manuscript (corresponding to Eq. (5) and (6) in the revised manuscript):
> > - Equation (6) (Equation (5) in the revised manuscript): $A_{i,t} \rightarrow \tilde{r}_{i,t}$
> > - Equation (7) (Equation (6) in the revised manuscript): $A\_{i,k} \rightarrow \tilde{r}\_{i,k}, \quad {\widetilde{A}}\_{i,t} \rightarrow \tilde{R}\_{i,t}$
> > - Equation (8) (Equation (7) in the revised manuscript): ${\widetilde{A}}\_{i,t} \rightarrow \tilde{R}\_{i,t}$
> >
> > ### **(2) The Remainder of the Manuscript**
> > We have replaced all instances of **"turn-level advantage"** with **"turn-level discounted return"** throughout the entire manuscript, including figures and appendices.
> >
> > --------
> >
> > ## **Response to Q3**
> >
> > We sincerely appreciate your suggestion regarding the exploration of a PPO-critic as a turn-level baseline estimator. While directly employing a PPO-critic currently presents challenges—specifically excessive memory and computational overheads, as well as the difficulty of aligning token-level granularity with turn-level steps—we agree that overcoming these obstacles to estimate the turn-level advantage (discounted return minus baseline) would enhance both the theoretical soundness and performance potential of our approach. Consequently, we are committed to actively exploring this direction in future work.
> >
> > ----------
> >
> > We appreciate your thoughtful suggestion to help us polish our paper. We will include these additional discussions and results in the final version. If you have any other questions, please feel free to tell us. Thank you once again!

---

### Meta-Review · Area_Chair_ko8o · 2026-01-08

**Summary:**

This submission introduced Information Gain-based Policy Optimization (IGPO), a reinforcement learning framework designed to address the challenges of sparse rewards in multi-turn tasks. IGPO provides dense, turn-level rewards by modeling each interaction as an incremental process of acquiring information about the ground truth, improving credit assignment and learning efficiency. The framework combines intrinsic, turn-level rewards with outcome-level supervision to create dense reward trajectories, significantly enhancing training for multi-turn tasks. Experimental results on both in-domain and out-of-domain benchmarks show that IGPO outperforms strong baselines, achieving higher accuracy and improved sample efficiency.

**Reviewer Concerns:**

The reviewers' concerns can be summarized into the following points:

1) Limited Applicability and Potential for Spurious Correlations: The proposed method is evaluated primarily on search-based QA tasks, where information-gain rewards are naturally aligned with task objectives. However, it remains unclear how this reward structure would perform in other domains like mathematical reasoning, code generation, or web agents. Additionally, the method may reinforce spurious correlations, as LLMs could exploit superficial cues rather than learning the underlying reasoning processes, which could undermine performance on tasks requiring more complex reasoning.

2) Insufficient Theoretical Justification and Novelty: While the paper introduces a new intrinsic reward design, it lacks robust theoretical support for this approach and does not sufficiently differentiate it from existing designs. The paper would benefit from a deeper discussion of alternative reward designs and a clearer exploration of how the proposed design can be integrated with other methods.

3) Concerns with Training Sensitivity and Computational Costs: The paper does not address the sensitivity of training to the normalization scheme of turn-level rewards, nor does it provide comparisons with other baseline methods like PPO. Additionally, the computational cost, particularly the need to compute per-turn log probabilities, is not discussed in sufficient detail, raising concerns about the scalability of the method.

**Reviewer Scores:**

Three reviewers respectively rated 4, 4, 6 respectively, and the second reviewer's review may not be a good reference, due to the LLM-generated manner (AC has checked the mismatch issues and consider that this is highly probably LLM generated contents). After the authors substantial rebuttal, AC thinks the first reviewer may raise their score, since the point-to-point questions have been well clarified and addressed during the author-reviewer discussion. For the third reviewer, his/her concerns about theoretical analysis, novel difference and alternative design has been carefully responded. AC has checked the theoretical proof and further clarification about the novelty and difference in design, and agreed with the authors' response. Overall, although the scores are below the acceptance rate, when removing the second reviewer's rating (more likely LLM generated), it seems the concerns raised by the other two reviewers have been well addressed by the authors.

---

### Decision · Program_Chairs · 2026-01-26

Accept (Poster)